# Artificial cysteine-lipases with high activity and altered catalytic mechanism created by laboratory evolution

Yixin Cen[1,2], Warispreet Singh [3,4], Mamatjan Arkin[1], Thomas S. Moody[4], Meilan Huang[3], Jiahai Zhou[2], Qi Wu[1] & Manfred T. Reetz[5,6]

Engineering artificial enzymes with high activity and catalytic mechanism different from naturally occurring enzymes is a challenge in protein design. For example, many attempts have been made to obtain active hydrolases by introducing a Ser → Cys exchange at the respective catalytic triads, but this generally induced a breakdown of activity. We now report that this long-standing dogma no longer pertains, provided additional mutations are introduced by directed evolution. By employing *Candida antarctica* lipase B (CALB) as the model enzyme with the Ser-His-Asp catalytic triad, a highly active cysteine-lipase having a Cys-His-Asp catalytic triad and additional mutations W104V/A281Y/A282Y/V149G can be evolved, showing a 40-fold higher catalytic efficiency than wild-type CALB in the hydrolysis of 4-nitrophenyl benzoate, and tolerating bulky substrates. Crystal structures, kinetics, MD simulations and QM/MM calculations reveal dynamic features and explain all results, including the preference of a two-step mechanism involving the zwitterionic pair $Cys105^-/His224^+$ rather than a concerted process.

[1] Department of Chemistry, Zhejiang University, 310027 Hangzhou, China. [2] State Key Laboratory of Bio-organic and Natural Products Chemistry, Shanghai Institute of Organic Chemistry, Chinese Academy of Sciences, 200032 Shanghai, China. [3] School of Chemistry and Chemical Engineering, Queen's University, Belfast, Northern Ireland BT9 5AG, UK. [4] Department of Biocatalysis and Isotope Chemistry, Almac Sciences, Craigavon, Northern Ireland BT63 5QD, UK. [5] Max-Planck-Institut für Kohlenforschung, 45470 Mülheim an der Ruhr, Germany. [6] Chemistry Department, Philipps-University, 35032 Marburg, Germany. Correspondence and requests for materials should be addressed to M.H. (email: m.huang@qub.ac.uk) or to J.Z. (email: jiahai@mail.sioc.ac.cn) or to Q.W. (email: wuqi1000@163.com) or to M.T.R. (email: reetz@mpi-muelheim.mpg.de)

Lipases and proteases are textbook examples of hydrolases which employ a key set of active-site residues for hydrolyzing esters and amides, respectively, and constitute one of the largest enzyme families in the human proteome[1,2]. Lipases function according to the classical Ser-His-Asp catalytic triad mechanism (Fig. 1a). Unlike lipases, proteases adopt divergent active sites and are grouped into seven mechanistic classes: serine-, cysteine-, aspartic-, metallo-, threonine-, glutamic-, and asparagine-proteases[3]. Among them, serine proteases have the Ser-His-Asp triad (Fig. 1b), while in the cysteine proteases, the catalytic triad consists of Cys-His-Asn (Fig. 1c)[4–7]. According to sequence alignment and phylogenetic analysis of proteases, the nucleophile exchanges Ser → Cys and Cys → Ser occurred during the evolution of both serine and cysteine proteases from common ancestors (Supplementary Fig. 1). Nevertheless, proteolytic active-site amino acids are the most evolutionarily conserved residues[8]. The structural and mechanistic similarities between lipases and serine proteases inspired us to raise the fundamental question whether the conserved serine-lipase can be evolved into an active cysteine-lipase by directed evolution.

In serine and cysteine proteases, investigators have studied the interconversion of serine and cysteine by either chemical[9–12] or recombinant technology[6,13,14]. In the classical studies, it was originally thought that oxygen and sulfur possess similar chemical properties, and that the exchange of these nucleophilic moieties would not influence catalysis notably. However, seminal experiments focusing on the generation of such mutant enzymes as Thiol-Subtilisin[9,10] and Thiol-Trypsin[11,13] demonstrated that these mutants are inactive toward common esters and amides, respectively. The reverse exchange in several cysteine-proteases, namely Cys → Ser, likewise causes notable reduction in activity (Supplementary Table 1). Similar results were found in the studies of other serine-hydrolases or cysteine-hydrolases (Supplementary Table 1). To the best of our knowledge, protein engineering of nucleophile-exchanged mutants with improved activity was never achieved.

For lipases[15,16], only a few reports of cysteine analogs have been reported, all resulting in greatly reduced enzyme activity (Supplementary Table 1). In contrast to native proteases, which possess either serine or cysteine nucleophilic residues, lipases are highly conserved with a serine nucleophile. It is therefore methodologically more challenging to engineer serine-lipases into highly active cysteine-lipases. In the present study we propose that the Ser → Cys conversion in lipases may lead to high activity by accurate manipulation of the local environment surrounding the hybrid Cys-His-Asp triad, enabled by laboratory evolution (Fig. 1d). We not only demonstrate this experimentally, but also provide an up to date theoretical analysis why the Ser to Cys exchange causes extreme activity reduction, while additional mutations result in high activity when testing substrates which are essentially not accepted by wild-type (WT).

Here, we choose lipase B from *Candida antarctica* (CALB), as the model enzyme, in which Asp187-His224-Ser105 is the catalytically active triad. As expected, Thiol-CALB (variant QW1; S105C) proves to be essentially inactive in the hydrolysis of such substrates as **1** and **4–19** (Fig. 2c). We succeed in evolving cysteine-CALB mutants that are even more active than WT CALB. Structural, kinetic and theoretical investigations point to a distinct catalytic mechanism, different from all serine-lipases known to date.

## Results

**Directed evolution of cysteine-CALB.** In order to examine the influence of the nucleophile exchange Ser → Cys on CALB activity, we selected ester **1** as the model substrate, which is hardly accepted by WT due to steric hindrance of the bulky benzyl group. Kinetic experiments using purified variant S105C (QW1) show that this mutation diminishes the catalytic efficiency by a factor of about two, with $k_{cat}/K_m$ (WT CALB) being 150 s$^{-1}$M$^{-1}$ compared to $k_{cat}/K_m = 88$ s$^{-1}$M$^{-1}$ for mutant QW1 (Table 1).

Protein engineering was pursued in order to achieve high activity of cysteine-CALB. Directed evolution and rational site-specific mutagenesis of enzymes are well known tools for improving their catalytic properties[17–19]. We used iterative saturation mutagenesis (ISM) at sites lining the binding pocket, a well established technique[17]. The choice of randomization sites was guided by the crystal structure of WT CALB (PDB code

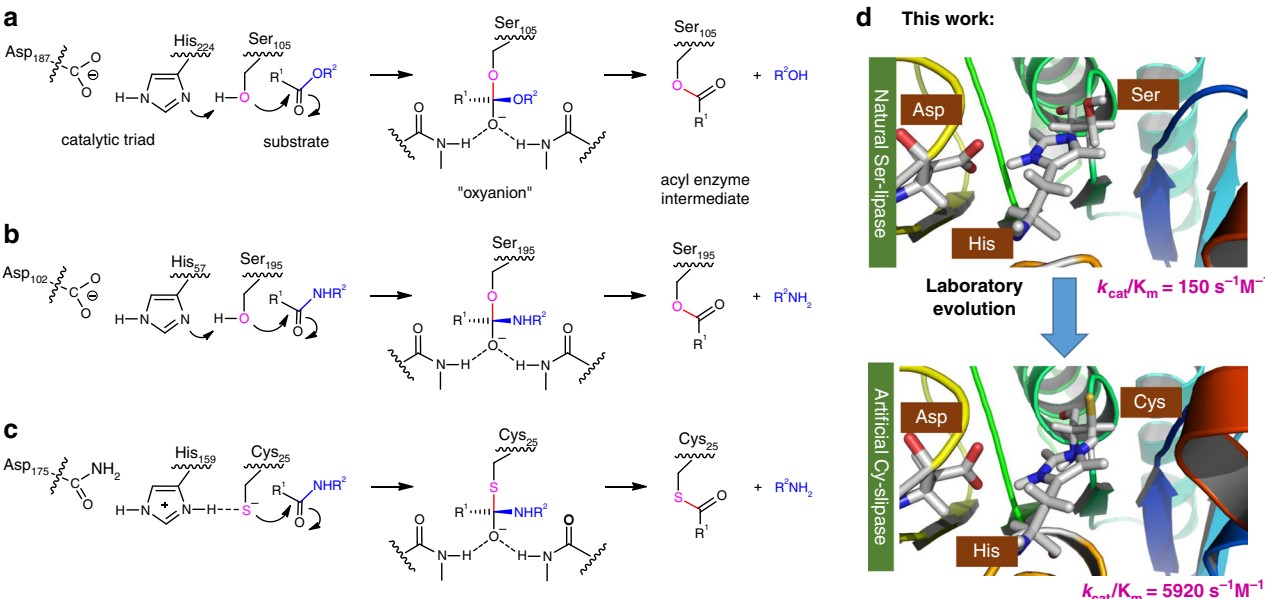

**Fig. 1** Schematic representation of the catalytic mechanism of natural hydrolases and artificial cys-lipase. **a** Lipases (CALB residue numbering). **b** Serine proteases (Trypsin residue numbering). **c** Cysteine proteases (Papain residues numbering). **d** Artificial cys-lipase evolved from natural ser-lipase (kinetic data referred to the hydrolysis reaction of model substrate **1**)

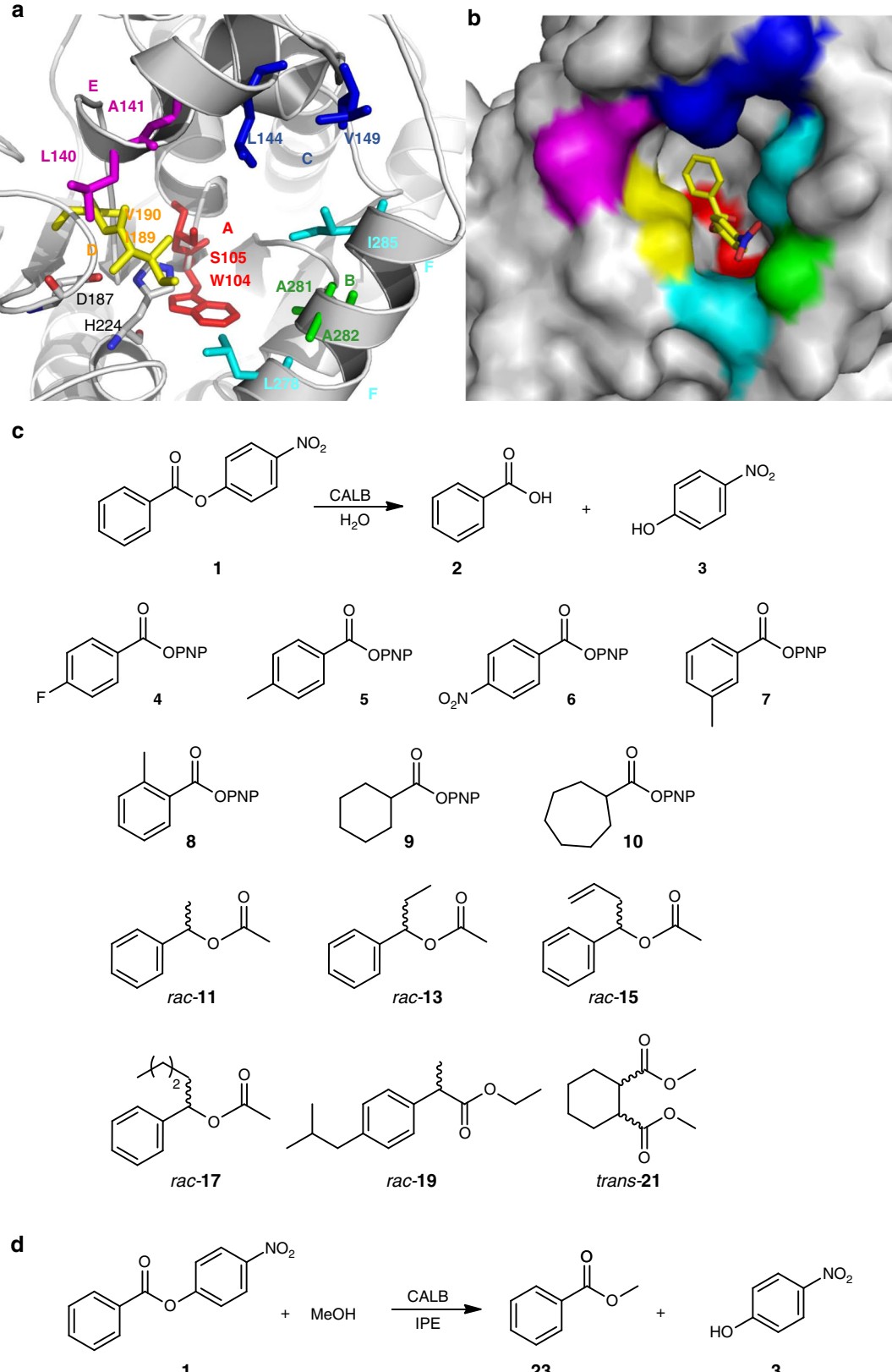

**Fig. 2** The choice of the ISM sites of CALB and enzymatic transformations of different substrates. **a** Selected 2-residue randomization sites A-F for ISM based on the X-ray structure of CALB:[21] Site A (W104/S105, red), B (A281/A282, green), C (L144/V149, blue), D (I189/V190, yellow), E (L140/A141, purple), and F (L278/I285, cyan). The catalytic triad Asp187-His224-Ser105 are shown in gray and red stick illustrations. **b** Binding pocket surface of WT CALB with the docked substrate **1**. Mutagenesis sites A, B, C, D, E and F are shown as surface illustration with the same color coding as in **a**. **c–d**, Structure and reactions of tested substrates

**Table 1 Kinetic data of WT CALB and mutants**

| Substrate | Entry | Enzymes | Mutations | $k_{cat}$ (s$^{-1}$) | $K_m$ (mM) | $k_{cat}/K_m$ (s$^{-1}$M$^{-1}$) |
|---|---|---|---|---|---|---|
| **1** | 1 | WT | – | 0.025 ± 0.009 | 0.17 ± 0.08 | 150 ± 11 |
| | 2 | QW1 | S105C | 0.028 ± 0.007 | 0.32 ± 0.02 | 88 ± 17 |
| | 3 | QW2 | W104V/S105C | 0.032 ± 0.005 | 0.50 ± 0.12 | 64 ± 6 |
| | 4 | QW3 | W104V/S105C/A281Y/A282Y | 0.031 ± 0.008 | 0.051 ± 0.02 | 610 ± 100 |
| | 5 | QW4 | W104V/S105C/A281Y/A282Y/V149G | 0.187 ± 0.024 | 0.032 ± 0.01 | 5920 ± 340 |
| | 6 | QW5 | W104V | 0.016 ± 0.003 | 0.014 ± 0.004 | 1187 ± 174 |
| | 7 | QW6 | V149G | 0.059 ± 0.009 | 0.077 ± 0.015 | 772 ± 48 |
| | 8 | QW7 | A281Y | 0.043 ± 0.011 | 0.066 ± 0.013 | 653 ± 100 |
| | 9 | QW8 | A282Y | 0.052 ± 0.014 | 0.513 ± 0.083 | 98 ± 16 |
| | 10 | QW9 | S105C/A281Y/A282Y/V149G | 0.041 ± 0.008 | 0.066 ± 0.011 | 610 ± 97 |
| | 11 | QW10 | W104V/A281Y/A282Y/V149G | 0.211 ± 0.012 | 0.043 ± 0.001 | 4914 ± 251 |
| | 12 | QW11 | W104V/S105C/A282Y/V149G | 0.027 ± 0.003 | 0.015 ± 0.004 | 1843 ± 313 |
| | 13 | QW12 | W104V/S105C/A281Y/V149G | 0.071 ± 0.006 | 0.031 ± 0.001 | 2256 ± 198 |
| **4** | 14 | WT | – | 0.030 ± 0.012 | 0.197 ± 0.093 | 150 ± 10 |
| | 15 | QW1 | S105C | 0.060 ± 0.003 | 0.134 ± 0.012 | 450 ± 18 |
| | 16 | QW4 | W104V/S105C/A281Y/A282Y/V149G | 1.605 ± 0.28 | 0.246 ± 0.059 | 6610 ± 510 |
| | 17 | QW10 | W104V/A281Y/A282Y/V149G | 2.609 ± 0.28 | 0.565 ± 0.070 | 4620 ± 70 |
| **5** | 18 | WT | – | 0.013 ± 0.001 | 0.018 ± 0.003 | 750 ± 140 |
| | 19 | QW1 | S105C | 0.008 ± 0.001 | 0.012 ± 0.003 | 660 ± 85 |
| | 20 | QW4 | W104V/S105C/A281Y/A282Y/V149G | 0.460 ± 0.026 | 0.075 ± 0.010 | 6140 ± 362 |
| | 21 | QW10 | W104V/A281Y/A282Y/V149G | 3.548 ± 0.583 | 0.424 ± 0.120 | 8426 ± 276 |
| **6** | 22 | WT | – | 0.022 ± 0.003 | 0.024 ± 0.018 | 1120 ± 320 |
| | 23 | QW1 | S105C | 0.042 ± 0.019 | 0.044 ± 0.025 | 990 ± 140 |
| | 24 | QW4 | W104V/S105C/A281Y/A282Y/V149G | 0.903 ± 0.179 | 0.138 ± 0.033 | 6550 ± 290 |
| | 25 | QW10 | W104V/A281Y/A282Y/V149G | 4.15 ± 0.96 | 0.630 ± 0.153 | 6600 ± 93 |
| **7** | 26 | WT | – | 0.099 ± 0.013 | 0.188 ± 0.021 | 523 ± 11 |
| | 27 | QW1 | S105C | 0.031 ± 0.004 | 0.066 ± 0.004 | 459 ± 25 |
| | 28 | QW4 | W104V/S105C/A281Y/A282Y/V149G | 0.165 ± 0.007 | 0.039 ± 0.008 | 4310 ± 346 |
| | 29 | QW10 | W104V/A281Y/A282Y/V149G | 0.307 ± 0.050 | 0.114 ± 0.014 | 2675 ± 115 |

*Note*: Source data are provided as a Source Data file

5A71)[20,21]. Twelve Amino acid residues surrounding the binding pocket were selected (Fig. 2a). In principle, we could start saturation mutagenesis using mutant QW1. However, in order to test whether there are any other possible nucleophile exchanges besides Ser → Cys in the WT, which could be discovered by directed evolution, residue Ser105 was also included in the saturation mutagenesis experiments. These 12 amino acid positions were then grouped into six randomization sites for subsequent ISM by combining 2 residues in a site (Fig. 2a, Sites A–F).

Saturation mutagenesis at site A was first tested starting from WT CALB. The use of NNK codon degeneracy encoding all 20 canonical amino acids would require in each case the screening of ≈3000 transformants for 95% library coverage[17]. In order to reduce the screening effort for 95% library coverage to about 500, we opted for NDT codon degeneracy encoding only 12 amino acids (F, L, I, V, Y, H, N, D, C, R, S, and G)[17]. Surprisingly, the screening result at site A showed that the most active variant in the reaction of substrate **1** was QW2 (W104V/S105C), a mutant of cysteine-CALB! The specific activity of QW2 showed a 2-fold improvement relative to WT (Supplementary Table 2).

We subsequently chose mutant QW2 (W104V/S105C) as a template for ISM at site B, which led to the identification of variant QW3 (W104V/S105C/A281Y/A282Y) with enhanced specific activity by a factor of 3.2 compared with WT CALB (8.7 μM × Min$^{-1}$ × OD$^{-1}$, Supplementary Table 2). The second-generation variant QW3 was then used in NDT-based mutagenesis at site C (L144/V149), which led to a series of variants with additionally improved activities (Supplementary Table 2). Among them, variant QW4 (W104V/S105C/A281Y/A282Y/V149G) showed the highest specific activity (11.4 μM × Min$^{-1}$ × OD$^{-1}$), which is 4.2-fold higher than WT CALB. Further ISM at sites D,

E, or F and screening based on the relatively crude on-plate pretest did not result in any improved mutants.

In order to accurately compare the catalytic profile of these variants, kinetic experiments were conducted using purified variants (Table 1, entries 1–5). QW2 ($k_{cat}/K_m$ = 64 s$^{-1}$M$^{-1}$) is slightly less efficient than QW1 or WT CALB. The apparent inconsistency between specific activities and catalytic kinetics is probably due to the crudeness of the screening pretest. Variant QW3 displays a catalytic efficiency ($k_{cat}/K_m$ = 610 s$^{-1}$M$^{-1}$) which is 9-fold higher than that of variant QW2. The subsequent ISM step resulted in a truly remarkable improvement. An extremely high catalytic efficiency was observed for the purified variant QW4 ($k_{cat}/K_m$ = 5920 s$^{-1}$M$^{-1}$), with $k_{cat}$ increased by 6 folds and catalytic efficiency ($k_{cat}/K_m$) increased by 40 folds compared with WT. Accordingly, in the time courses of CALB-catalyzed hydrolysis of **1**, 90% conversion was achieved by variant QW4 in only 2 h, while the conversion was less than 10% for WT (Supplementary Fig. 3a). This implies the successful construction of a highly active cysteine-lipase with a hybrid catalytic triad Cys-His-Asp.

**Deconvolution experiments**. In order to explore the effect of each amino acid exchange in the best variant QW4 on the significantly enhanced activity in the hydrolysis of ester **1**, partial deconvolution was performed by generating four multiple-point variants and four single point variants (Table 1, entries 6–13). Compared with WT, three single point variants (QW5, QW6, and QW7) showed improved catalytic profiles (Table 1, entries 6–8), implying the importance of the W104V, V149G, and A281Y mutations. Their effects can also be seen from the remarkably reduced activity of corresponding multiple-point mutants (QW9,

QW3, and QW11) when compared with the best variant QW4 (Table 1, entries 10, 4, 12). Activity assessments of these single point variants and multiple-point variants indicate that the effect of these mutations is cooperative rather than additive.

Notably, variant QW10, which has the natural catalytic triad Ser-His-Asp restored and only differs from the best variant QW4 by the residue at position 105, displays lower activity than QW4 with the hybrid Cys-His-Asp triad (Table 1, entry 11). This clearly implies the feasibility of creating a highly active cysteine-lipase with the hybrid Cys-His-Asp triad by means of directed evolution.

**Substrate scope**. A series of other bulky esters were also tested as expanded substrates, because most of them are not well accepted by WT CALB due to steric hindrance (Table 1, entries 14–29). The kinetic data shows, inter alia, that variants QW4 and QW10 are much more active for several substituted benzoates than WT and QW1. When comparing kinetic data or reaction time course, QW4 is more efficient than QW10 in the conversion of **4** and **7**, and almost as active as QW10 for **5** and **6** (Table 1, Supplementary Fig. 3). Other sterically hindered substrates such as *p*-nitrophenyl cyclohexanecarboxylate (**9**) or *p*-nitrophenyl cyclo-heptanecarboxylate (**10**) are also efficiently hydrolyzed by QW4 and QW10 (Supplementary Fig. 3f–g).

WT CALB is an excellent catalyst in the hydrolytic kinetic resolution (KR) of appropriate esters of chiral secondary alcohols with preferential formation of (*R*)-enantiomers[22]. The present mutants were tested in the reactions of four *sec*-alcohol esters, *rac*-**11** ~*rac*-**17**. As expected, WT showed good selectivity and activity for *rac*-**11** ($E > 200$ (*R*), Supplementary Table 3, entry 1), *rac*-**13** ($E = 71$ (*R*), Supplementary Table 3, entry 6) and *rac*-**15** ($E = 83$ (*R*), Table 2, entry 1), while QW1 showed reduced activity and low enantioselectivity. Upon using the best mutant QW4, notably improved conversion was observed compared with QW1 (Table 2, Supplementary Table 3). Remarkably, QW4 induces reversed enantioselectivity in favor of the (*S*)-alcohol, similar to variant W104A previously reported by Hult et al.[22]. In the case of *rac*-**15**, variant QW4 is superior to QW10 (Table 2), and for *rac*-**13** they are similar, while for *rac*-**11** QW10 is a little

better (Supplementary Table 3). It is noteworthy that *rac*-**17** with a large alkyl group also can be accepted by QW4 and QW10 with moderate selectivity, in sharp contrast to WT CALB which has no activity toward this substrate due to the limited space of the alcohol-binding pocket (Supplementary Table 3, entry 11–14). Moreover, WT CALB is also a poor catalyst in the KR of *rac*-**19** with the stereogenic center in the acid fragment, and low activity and poor enantioselectivity were observed ($E = 2$ (*R*), Supplementary Table 3, entry 15). Thiol-CALB and other mutants also perform poorly for this type of esters (Supplementary Table 3, entry 16–18), which outlines the limitation of the CALB mutants. In the case of KR of *trans*-**21** (trans-dimethyl cyclohexane-1,2-dicarboxylate), surprisingly, QW1 (Thiol-CALB) displays unexpectedly excellent performance for this specific substrate like WT CALB ($E > 200$ (*R*), Supplementary Table 3, entry 19–20). However, both QW4 and QW10 variants showed very low activity for *trans*-**21**, probably because their reshaped active sites cannot accept this substrate well. Interestingly, high stereoselectivity for *trans*-**21** with ee values up to 99% was also observed for this reaction catalyzed by QW4 and QW10 variants, respectively (Supplementary Table 3, entry 21–22).

We further tested the nonaqueous transesterification of the model substrate (**1**) using methanol catalysed by WT CALB and QW4, QW10 variants. The results clearly demonstrate excellent performance of the cysteine-lipase QW4 for this bulky substrate, in distinct contrast to WT CALB (Supplementary Fig. 4).

**Crystal structure characterization**. In order to gain insight into the basis of increased activities of QW4 and QW10, the crystal structures of both enzymes were solved and refined to 1.85 and 1.88 Å resolution, respectively. Surprisingly, Cys105 in variant QW4 appeared to be oxidized to the respective sulfinic acid. When purification and crystallization were performed in a glove box, Cys105 in the crystal structure was in the normal (reduced) state. In fact, cysteine proteases are readily oxidized[23–27], and to date reduced cysteine has only been observed in a few cases such as the crystal structures of inactive thiol proteases derived from serine proteases[13,27] or non-catalytic free cysteine[24,28], which implies that the active site sulfur in QW4 is reactive.

Most residues in the structures of QW4 and QW10 have very similar conformations. The most remarkable difference was observed at loop 137–150 and helix 277–288 (α-10 helix[21]). Loop 137–150 of QW4 and QW10, the lid at the entrance of catalytic sites in CALB[21], is very flexible as compared to those in WT. This was indicated by higher B-factors (Fig. 3a–c) and RMSF in subsequent molecular dynamics (MD) simulation (Supplementary Fig. 8). The raised mobility of loop 137–150 in QW4 and QW10 enables ready entry of bulky substrates[29–31], which can be attributed to the introduction of the V149G mutation. In addition, there are different H-bond networks between the α-6 helix (residues 151–157), loop 137–150 and α-10 helix (residues 267 to 288) in QW4, QW10, and WT-CALB (Fig. 3g–i). In QW10 and WT, three or four H-bonds exist between S150/W155 and Q291 (Fig. 3g, i), while in QW4 there are only two H-bonds between G149/W155 and Q291 (Fig. 3h). These differences in the H-bond interactions result in a more flexible and wider open lid in QW4 than those in QW10 and WT (Fig. 3h, e).

In addition, in the crystal structure of QW4, the B-factors of the α-10 helix (residues 267–288), especially at helix 277–288, are much higher than those in WT and QW10 (Fig. 3a versus 3b–c). This increased flexibility may aid the substrate to enter the binding pocket and to bind at the active site, thus leading to the high activity of QW4 in the hydrolysis of substrate **1**. Interestingly, the nucleophile exchange (S105C) in QW4 results in a conformational change of helix 277–288, compared with

**Table 2 Stereoselectivity of hydrolytic kinetic resolution of *rac*-15**

| Entry | Enzyme | Reaction time (h) | Conv. (%) | ee_p (%) | E |
|-------|--------|-------------------|-----------|----------|---|
| 1 | WT | 24 | 49 | 93 | 83 (R) |
| 2 | QW1 | 36 | 6 | 56 | 4 (R) |
| 3 | QW4 | 36 | 31 | 93 | 41 (S) |
| 4 | QW10 | 36 | 18 | 93 | 33 (S) |
| 5 | W104A | 24 | 48 | 82 | 23 (S) |

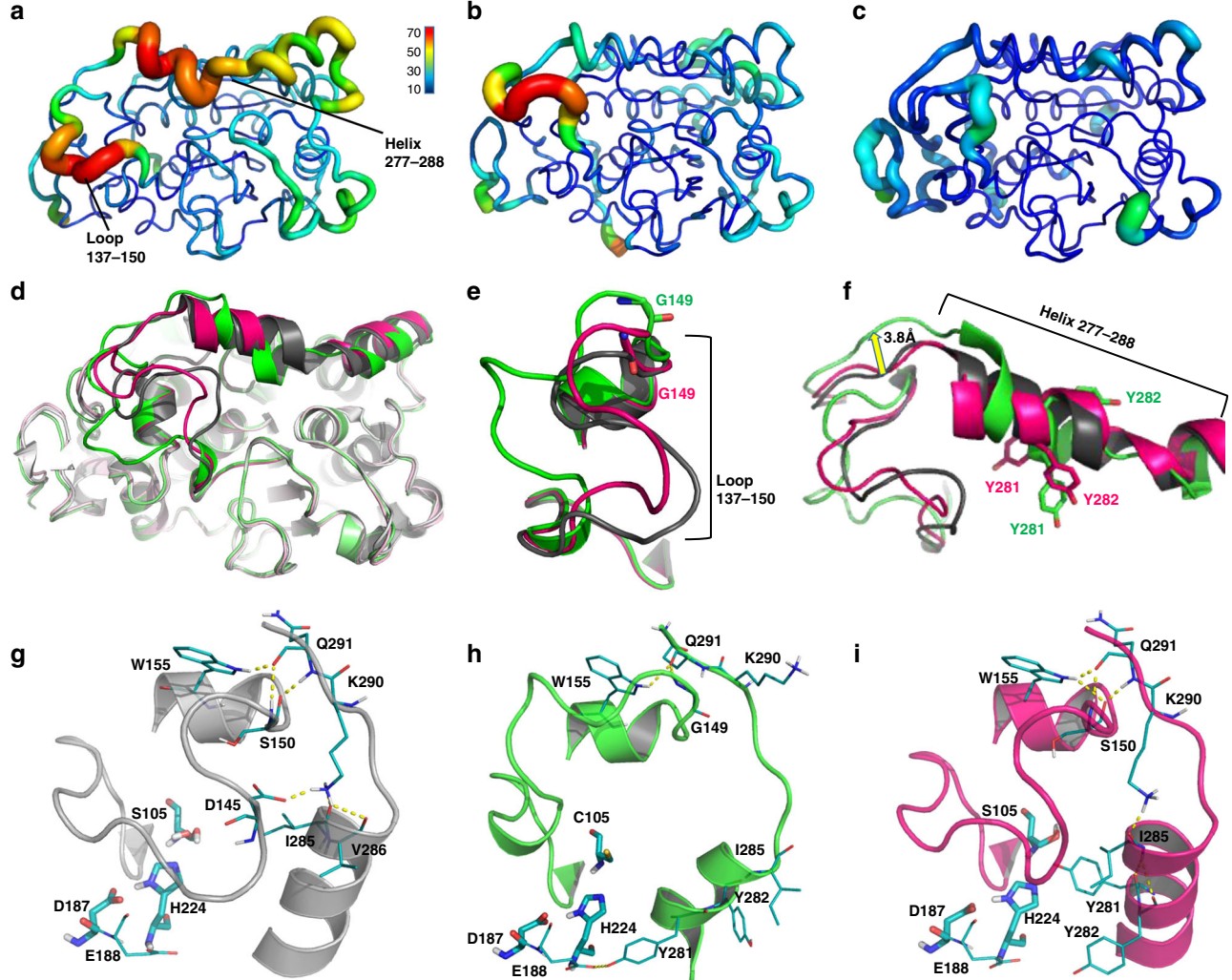

**Fig. 3** Cartoon representation of the crystal structures of enzymes. **a** Variant QW4. **b** Variant QW10. **c** WT CALB (PDB: 5a71[21]). **a–c** are colored according to the B-factor. **d** Superimposed overall structures of WT CALB (gray), QW4 (green) and QW10 (pink). **e** Zoom-in view of the loop 137–150 in WT CALB (gray), in QW4 (green) and in QW10 (pink). **f** Zoom-in view of helix 277–288 in WT CALB (gray), in QW4 (green) and in QW10 (pink). Hydrogen bonds formed between the α-6 helix, loop 137–150 and α-10 helix in **g** (WT-CALB), **h** (QW4) and **i** (QW10). H-bond formed between Tyr281 and Glu188 in QW4 mutant (**h**) was also noted

QW10 or WT. While the helixes of QW10 and WT CALB are well superimposable (Fig. 3f), the helix section P280-Y282 of QW4 is deformed and shifted toward the active site, with an additional H-bond between Y281 and E188 formed (Fig. 3h). This conformational change also induces loop 289–294 next to helix 277–288 to shift by about 3.8 Å towards the solvent. This effect leads to more room for the substrate at the entrance to the binding pocket (Fig. 3f). By comparing the kinetic data of QW3 and QW2, we found $K_m$ is reduced by a factor of 10 (Table 1, entries 3–4). This indicates that the above noted movement of helix 277–288 is likely to be present in QW4, where the A281Y/A282Y mutation is also introduced.

**Substrate binding disclosed by MD simulations.** In order to interpret more precisely the differences in catalytic activities of these enzymes in presence of a substrate, p-nitrophenyl benzoate (**1**) was docked into the binding pockets of WT-CALB, and variants QW2, QW4, and QW10, respectively, and 100-ns MD simulations were run. The results revealed distinctly different modes of binding in WT and in variants QW4 and QW10 (Fig. 4). It was observed that W104 in WT clearly hinders access

of the substrate into the proximity of the active site and results in a bent conformation of **1** (Fig. 4a, Supplementary Figs. 7a and 9).

For QW10 and QW4 (Fig. 4c, d), S105 and C105 remain in close proximity to the substrate carbonyl group, favorable for nucleophilic attack to occur (Supplementary Fig. 7d-e). Compared with QW2 (Fig. 4b), introducing the A281Y/A282Y mutations in QW4 and QW10 causes obvious displacement around the helix region 279–289. P280 and Y281 move in the proximity to the substrate, and form favorable hydrophobic stacking interaction with the phenyl ring of **1**. The loop following the helix 279–289 moves accordingly, so that the H-bond between K290 and V286 observed in WT and QW2 is broken, and a new H-bond interaction is established between G149 and Q291, resulting in an open conformation of the lid. Extra space is created by the induced open conformation of the QW4 and QW10 variants (Fig. 4g, h) compared with WT and QW2 (Fig. 4e, f), which may also account for the high activities of the best mutants. The interactions with Y281 and E188 in QW4 further positions the substrate in the catalytic pocket of the enzyme. Moreover, the presence of a water molecule, which forms an

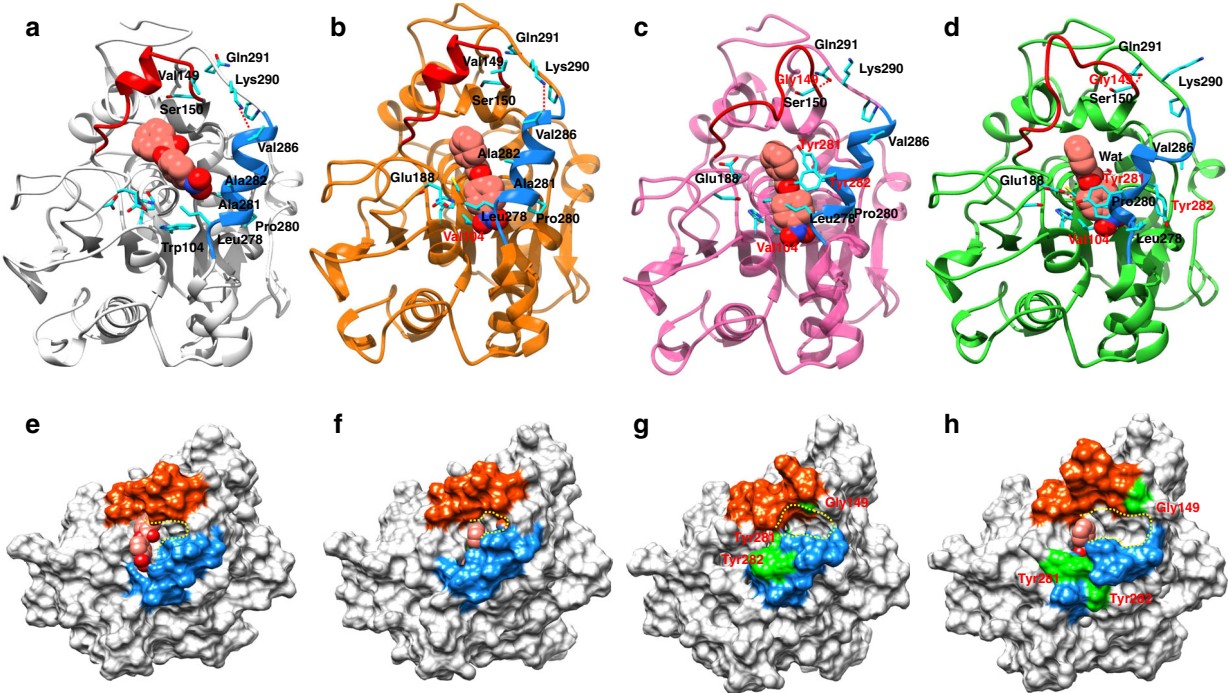

**Fig. 4** The binding modes and surface representation of substrate **1** in CALB and its mutants. The structures were obtained from cluster analysis of the 100-ns MD trajectory. **a** and **e**, WT CALB after 15 ns of MD simulation. **b** and **f**, Variant QW2. **c** and **g**, Variant QW10. **d** and **h**, Variant QW4. Loop 137–150 is shown in red cartoon and surface. Helix 277–288 is shown in blue cartoon and surface. Substrate **1** is shown as a red sphere

H-bond with substrate **1** in QW4 (Fig. 4d), may be critical for the subsequent hydrolysis reaction, and therefore also be responsible for its high catalytic efficiency.

**Altered reaction mechanism of cysteine-CALB variant QW4.** In order to compare the reaction processes and mechanistic details of lipases with naturally occurring Ser-His-Asp triad versus engineered Cys-His-Asp triad on a molecular level, we applied QM/MM[32,33]. Previous QM/MM studies on cysteine proteases, including human Cathepsin K and papain featuring a Cys-His-Asn triad, indicated that the catalytic Cys is deprotonated prior to nucleophilic attack in the formation process of acyl-enzyme complexes[34,35]. In contrast to papain-like cysteine protease, the QM/MM study on another cysteine protease such as legumain revealed that the catalytic Cys remains protonated during nucleophilic attack, and a concerted mechanism of simultaneous proton transfer and nucleophilic attack was suggested[36]. QM/MM studies on the formation process of acyl-enzyme complexes of serine proteases also disclosed distinct mechanisms[37,38]. In view of the complexity of the mechanism of proteases, we propose that the cysteine-lipase mutant QW4 with the hybrid Cys-His-Asp triad catalyzes the acylation reaction via a mechanism distinctly different from CALB with the naturally occurring triad Ser-His-Asp.

An extensive QM/MM investigation showed that for the QW10 variant, the first step of the formation process of acyl-enzyme complexes works via a concerted mechanism in which the proton transfer from S105 to H224 occurs simultaneously with the nucleophilic attack to the carbonyl carbon atom of the substrate (Fig. 5a, Supplementary Fig. 10). The calculated reaction barrier is 10.56 kcal/mol and the reaction is exothermic, the energy of the tetrahedral product being 25.75 kcal/mol lower than that of the reactants. We have also considered the possibility of a step-wise mechanism for QW10. The stable structure of the ionic pair form of the respective zwitterionic ion could not be obtained from scanning the reaction coordinates corresponding to the proton

transfer from the Ser105 hydroxyl to the His224 Nε. Any attempt to optimize the geometry closest to the ionic pair form from the potential energy surface always led to the neutral form. In addition, the energy kept increasing with the decrease of the reaction coordinate, and no transition state corresponding to the proton transfer from the Ser105 hydroxyl to the His224 Nε could be located (Fig. 5a, Supplementary Fig. 11). This indicates that the reaction proceeds according to the concerted mechanism instead of the step-wise alternative.

In striking contrast, the potential energy surface scan for QW4 disclosed that the acylation reaction follows a two-step mechanism involving a HisH⁺/CysS⁻ ion pair, in which the proton is first transferred from the thiol group of C105 to H224, and then the negatively charged thiolate anion acts as a nucleophile, attacking the carbon of ester bond of the substrate to yield the tetrahedral intermediate. The first step, i.e., the proton transfer from SH of C105 to H224 is the rate determining step with an activation barrier of 13.95 kcal/mol, and then the thiolate anion attacks the carbonyl of the substrate rapidly with a low barrier of 6.2 kcal/mol (Fig. 5b, Supplementary Fig. 12).

We then posed the question whether the concerted mechanism is a viable alternative for QW4. Thus we scanned the reaction coordinates between the thiol sulfur of C105 and the substrate carbonyl carbon, in order to obtain a transition state corresponding to the simultaneous proton transfer from C105 to H224 and nucleophilic attack of C105 thiol group to substrate carbonyl (Supplementary Fig. 13). However, optimization of the point which is closest to a transition state corresponding to the concerted mechanism always led to the Cys105⁻/His224⁺ zwitterioinic pair, indicating the concerted mechanism is not preferred for variant QW4.

In further work, we conducted QM cluster calculations[32,33,39], hoping to validate the two-step reaction mechanism proposed for variant QW4. Starting from the ionic pair, the estimated energy barrier and transition state are in agreement with the QM/MM results (Supplementary Fig. 14).

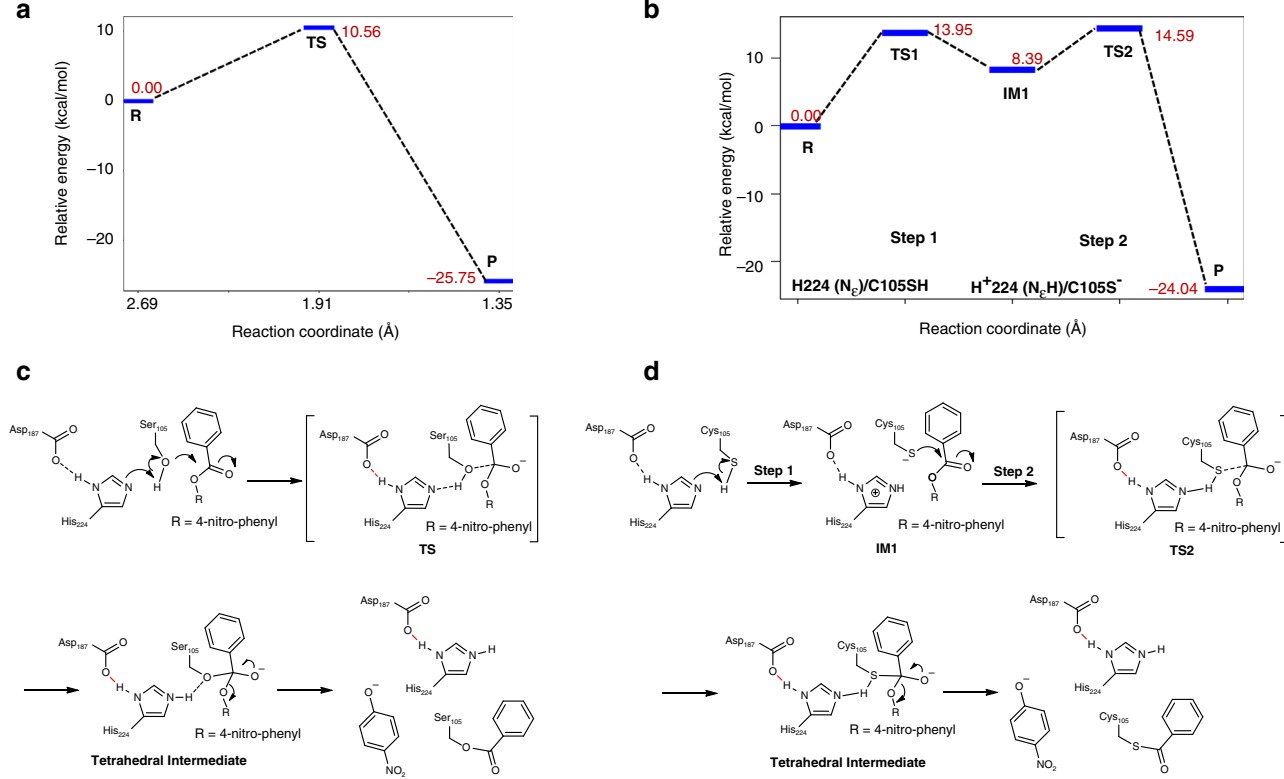

**Fig. 5** Different reaction profiles of the normal CALB variant QW10 and cysteine-CALB variant QW4. **a** and **c**, The normal CALB variant QW10 adopts a concerted mechanism of simultaneous proton transfer and nucleophilic attack. **b** and **d**, For the cysteine-CALB variant QW4, the formation of the tetrahedral intermediate occurs via a two-step reaction mechanism where proton transfer from Cys105 to His224 is followed by a nucleophilic attack of the deprotonated Cys105 to the carbonyl group of the substrate. The reaction profiles in the hydrolysis of substrate **1** were calculated by QM/MM (AMBER/ωB97X-D/6-31 G + (d, p)). The reaction coordinate is the distance between the substrate carbonyl carbon and hydroxyl oxygen of Ser105 or thiolate sulfur of deprotonated Cys105

The concerted mechanism was also examined for QW4 using the QM cluster model. Again, no transition state corresponding to a concerted mechanism was located (Supplementary Fig. 15).

## Discussion

It is known that the overall topologies and catalytic properties of most serine and cysteine hydrolases are remarkably different, indicating that they evolved separately as a result of convergent evolution. On the other hand, nucleophile exchanges of this kind do occur in the evolutionary history[8] of serine and cysteine proteases. For example, viral cysteine proteases appear to be homologous to the trypsin family of eukaryotic serine proteases[40,41]. These cases indicate that the interconversion of nucleophilic serine and cysteine is in principle possible, while the local environment around the distinct catalytic triads exists in an optimized form favouring respective processes during natural evolution due to the differences in pKa, nucleophilicity and the leaving group properties of oxygen versus sulfur in these enzymes[5]. Indeed, many experiments focusing on the simple exchange of nucleophilic serine and cysteine cause notable reduction in activity[9–14].

To date, protein engineering of nucleophile-exchanged mutants with improved activity has never been achieved. In our study, the change from Ser to Cys at the catalytic triad of CALB diminishes catalytic efficiency (Table 1) as in all previous reports of other lipases which had undergone such a mutational switch (Supplementary Table 1). Directed evolution of cysteine-CALB with generation of mutant QW4 led to a distinct improvement of activity for a range of bulky substrates, which are not well

accepted by WT CALB. Our results show that lipases can indeed tolerate the nucleophile exchange from serine to cysteine in the Cys-His-Asp catalytic triad, which actually enables otherwise difficult transformations.

The results of deconvoluting the best cysteine-CALB variant QW4 clearly demonstrate pronounced cooperative effects of the five mutations in this enzyme, which we attribute to the significant changes in the CALB structure. These epistatic effects were also confirmed by X-ray structure analysis and MD simulations. It is remarkable how the introduction of W104V, V149G and A281Y/A282Y mutations reshapes the structure of the catalytic active site in the QW4 variant, including (1) the enlarged space at the active site enabling an optimal degree of binding of bulky substrates; (2) the increase in flexibility of the α5 lid 135–150 and of the helix 277–288, enabling ready opening of the lid and the substrate to be easily accepted; (3) the formation of favorable hydrophobic stacking interaction between Y281, P280 and the phenyl ring of **1**, caused by the H-bond between E188 and Y281 and the notable displacement around the helix region 279–289.

Modern QM/MM techniques not only allow for the elucidation of mechanistic details of enzymes, they can also be used to predict enzyme reaction mechanisms[32,33]. In contrast to the widely studied reaction mechanism of proteases[34–38], little information is available concerning the mechanism of lipases based on QM/MM calculations[42–44]. On the basis of X-ray structural data and QM/MM calculations, we observed the difference in reaction mechanism between CALB with the naturally occurring triad of Ser-His-Asp and the cysteine-CALB mutant (QW4) with a Cys-His-Asp triad. The high catalytic efficiency of QW4 is attributed

to its optimal binding properties, as well as the relatively low reaction barrier in the rate-determining step.

In summary, the successful construction of a highly active cysteine-lipase mutant with hybrid Cys-His-Asp triad expands the types of lipases that can be discovered and be functionally active. Our results suggest that besides the protease enzyme family, the occurrence of a Ser to Cys nucleophile exchange in the lipase family is also possible, in this case by laboratory evolution. This raises the intriguing question whether cysteine-lipases can be found in nature. The present work provides a valuable insight into the catalytic mechanisms of naturally occurring lipases and artificial cysteine-lipases at the molecular level and how these catalysts can transform structurally different substrates which are not accepted by the WT lipase. This opens up exciting new opportunities for these engineered lipases in chemical synthesis.

## Methods

**Library screening and kinetic measurements**. Saturation mutagenesis libraries were constructed at sites A (Trp104/Ser105), B (Ala281/Ala282), C (Leu144/Val149), D (Ile189/Val190), E (Leu140/Ala141), and F (Leu278/Ile285). PCRs were performed using WT-CALB plasmid (pETM11-CALB) as the template DNA, and forward primers (see Supplementary Table 5) and a silent reverse primer (GGGAGCAGACAAGCCCGTCAGGG, 2444−2466 bp of pETM11). The reaction (100 μL final volume) contained: 10 × KOD buffer (10 μL), MgCl₂ (4 μL, 25 mM), dNTP (10 μL, 2 mM each), forward primers (4 μL, 2.5 μM each), silent reverse primer (4 μL, 2.5 μM each), template plasmid (1 μL, 100 ng μL⁻¹) and 1 μL of KOD polymerase. PCR conditions used were 95 °C, 3 min; five cycles of (98 °C, 1 min; 65 °C, 1 min; 72 °C, 5 min) for the generation of megaprimer; 20 cycles of (98 °C, 1 min; 72 °C, 8 min); and final extension at 72 °C, 10 min[45]. The purified PCR products were transformed into electrocompetent cells of *E. coli* BL21 (DE3) (containing chaperone plasmid pGro7, Takara, Japan). Transformants grown on LB-agar plates containing Kanamycin (34 μg/mL) and chloramphenicol (34 μg/mL) were picked up and cultured overnight at 37 °C with shaking (800 rpm) in 96 deep well plates containing 800 μL TB with Kanamycin (34 μg/mL) and chloramphenicol (34 μg/mL). After inoculation into fresh TB media containing antibiotics and L-arabinose (1 mg/mL) for 4 h at 37 °C, isopropyl β-thiogalactopyranoside (IPTG) (final concentration 1 mM) was added to induce the expression of CALB. After expression for 24 h at 25 °C, cells were harvested by centrifugation at 2750 × g and 4 °C for 25 min., and treated with lysozyme and Dnase I. The supernatants of CALB libraries were transferred to microtiter plates for screening using a UV/Vis-plate reader.

The kinetics data of purified enzymes as catalysts were determined on a UV/Vis-plate reader by monitoring the time-dependent appearance of *p*-nitrophenolate (**3**) in the hydrolysis reactions of substrate **1**, **4–7** at various concentration ranges. The obtained data were fitted to the Michaelis-Menten equation by nonlinear regression analysis.

**Crystallization and structural determination**. QW4 and QW10 were crystalized by using the sitting-drop vapor diffusion method at 18 °C. All crystals were mounted in nylon loops and flash-frozen in liquid nitrogen. Diffraction data of QW4 (oxidized), QW4 (unoxidized) and QW10 were collected on SSRF beamline of the National Center for Protein Science Shanghai (China). The data set were indexed, integrated, and scaled using HKL3000[46] or XDS[47] package. All structures were solved by molecular replacement method with PHASER[48], and refined with PHENIX[49] and COOT[50]. The statistics for data collection and crystallographic refinement are summarized in Supplementary Table 4. Further details can be found in Supplementary Methods.

**Computational methods**. X-ray structural data of PDB accession code 6ISR and 6ISP in this paper, and the published 5A71[21] were used as the starting points for mutants QW4, QW10 and WT CALB, respectively. The input PDB file of variants QW1 was constructed basing on WT CALB (5A71.pdb) by using PyMOL.99rc6 program[51]. The MD simulations were performed with Amber 14 software[52]. Average structures were obtained from the conformers of 100 ns of MD trajectory after 1 ns of equilibration. The docking process was performed by using Autodock 4.0[53]. The energetically favorable poses of the substrates **1** binding to the targeted binding site of WT CALB, variants QW1, QW10, and QW4 were extracted. The snapshots for the QM/MM calculations obtained from cluster analysis of the 100 ns MD trajectory were subjected to energy minimization by 5000-step steepest descend and 5000-step conjugate gradient algorithms. The enzyme-substrate complex together with the water shell of 8 Å surrounding the enzyme was included in the QM/MM calculations. All the calculations were run using DFT with B3LYP/ωB97X-D functional and 6–31+G (d, p) basis set implemented in Gaussian 09[54].

**Reporting summary**. Further information on research design is available in the Nature Research Reporting Summary linked to this article.

## Data availability

The data that support the findings of this study are available from the corresponding author upon reasonable request. The crystal structure of mutants QW4 (oxidized), QW4 (unoxidized) and QW10 has been deposited in the Protein Data Bank (PDB) under accession code 6ISQ, 6ISR, and 6ISP, respectively. The raw data underlying Table 1 and Supplementary Figs. 2–4, 11, 13, 15 and Supplementary Table 2 are provided as a Source Data file.

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

## Acknowledgements
This research was funded by the Max-Planck-Society and the Arthur C. Cope Fund (to M.T.R.), and by National Natural Science Foundation of China (No. 21574113) and Zhejiang Provincial Natural Science Foundation (No. LY19B020014) (to Q.W.), and by INVEST NI Research and Development Program (to M.H. and T.S.M), and by the Strategic Priority Research Program (B) of the Chinese Academy of Sciences (XDB20000000 to J.Z.). We thank the staffs from BL17U, BL18U, and BL19U1 at Shanghai Synchrotron Radiation Facility (SSRF, China) for assistance during X-ray data collection.

## Author contributions
Q.W., Y.C. and M.A. performed the experiments; W.S., T.S.M and M.H. performed the MD simulations and QM/MM study; Y.C. and J.Z. performed the crystallization of mutants; the project was designed and supervised by Q.W. and M.T.R.; M.T.R., Q.W. and M.H. wrote the paper; all authors checked the paper.

## Additional information

**Competing interests:** The authors declare no competing interests.

