## [Peer Review File · Nature Communications]

Reviewers' comments:

Reviewer #1 (Remarks to the Author):

Authors describe the preparation of different new artificial calB variants by directed evolution, focus on the modification on particular positions on the structure.

They found some variants with better improvement in activity and some results in inversion in enzyme enantioselectivity. However, the main point in the idea which is the cysteine-lipases is not clear to be such relevant

They found activity in the lipase after mutation catalytic serine by cysteine, conserving around 50% the initial activity, however the final better results in the new artificial enzyme is based on other modifications, W104, A281, A282 and V149.

Indeed, the QW10 variant, which contain these 4 mutations conserving intact the catalytic triad is much better in general than the cysteine 105 containing variant: Hydrolytic activity of several substrates identical, and much better for QW10 with 5 and quite similar with substrate 1.

Furthermore, a critical point is the selectivity where in the reaction in table 2, both QW10 and QW4 showed the same inversion in enantioselectivity, moderate value, but with substrate 11, an E value of 63S for QW10 against 13S for QW4 (also the reaction was faster for QW10). Also for rac-15 QW10 was better.

It has been well described that W104 has a critical role in the enantioselectivity of this enzyme. Even authors previously published critical changes in selectivity for this enzyme in JACS, 2013, and other authors (Hult, *chembiochem*, 2010, OBC, 2011). Also they employed similar amino acid positions such as 282 in combination with W104 for variants showing better activity and selectivities.

Therefore, the results seem demonstrate that the relevant alterations in the better variants are W104, A282 and A281 (oxyanion) instead of the modification of the catalytic serine, which is harmful in many cases.

Most studies in this work have been performed with model substrate and also it would be important testing the variants, specially WQ10, in other interesting reactions where high alteration in selectivity, enantio or preference can be demonstrated.

Reviewer #2 (Remarks to the Author):

Comments to:

Cen et al. – Artificial cysteine-lipases with high activity and altered catalytic mechanism created by laboratory evolution.

This paper reports on the engineering of CALB into a "cysteine lipase". This is the first time that a lipase is successfully mutated from a serine lipase into a cysteine lipase, and it works a proof of concept, influencing thinking in the field. The work is technically sound and well written. The only concerns come from the points below, which should be addressed before publication.

Fig. S1 and introduction. In the manuscript introduction, authors refer to the proteases alignment in fig. S1 as phylogenetic analysis of proteases. A phylogenetic analysis requires a more consistent amount of data to be compared and, moreover, a detailed phylogenetic tree based on a conserved protein domain, on the aminoacidic stretch containing the active site or on the full protein/enzyme sequence. The alignment of four sequences is indicative to show the Cys to Ser exchange between different organisms but not enough to support the concept of phylogeny.

*Page 5. Crystal Structure characterization. The refinement resolution indicated for QW10 is 2.1 Å. In the supplementary table S4 is indicated however a resolution of 1.72 Å for data collection with reasonable data statistics. What is the reason to cut back the refinement resolution to 2.1 Å?

*Supporting Information. Crystallization and X-ray Structural Analysis. Second last line. Supplementary Table S3 should be S4 instead.

*Supporting Information. Supplementary Figure S4. Please indicate the sigma level used to contour the map.

*Supporting Information. Supplementary Table S4. The data collection statistic should be indicated for the resolution used for refinement ie. 2.1 Å of QW10. Please present data for the last resolution bin too.

*Supporting Information. Supplementary Table S4. Data collection. Please indicate Wilson B-factor.

*Supporting Information. Supplementary Table S4. Refinement. % Ramachandran outliers, % Ramachandran favored, % rotamer outliers, should be indicated.

*Supporting Information. Supplementary Table S4. The Rfree/Rfactor are considerably high for QW10 when compared to QW4 in both forms. Also the number of ligands (401) and waters (442) is high for QW10 when compared to QW4 in both forms, given also the different resolutions. It appears that there are some problems with the refinement of this structure. What is like the plot of Rfree/Rfactor vs resolution for QW10? How was the space group determined?

*Results. Authors state that the inconsistency between specific activities and catalytic kinetics between QW1 and WT CALB is due to the different expression levels of the mutants. In order to obtain comparable kinetics data, equal amounts of purified variants should be the starting point for the subsequent generation of data. The equal amount of protein is not correlated to the expression level of the mutant but it is a downstream normalization. Moreover, Authors compare the catalytic efficiency of the variants QW2, QW3, QW4. If true what previously mentioned, also in this case the expression level should be considered and the starting amount of proteins equalized to have comparable data.

*Reference #10. "subtillisin" should be "subtilisin"

*Reference #14. "viology" should be "virology"

Reviewer #3 (Remarks to the Author):

The authors have engineered highly active lipases for bulky substrates, employing *Candida antarctica* lipase B (CALB), notably some of which have Ser replaced by Cys in the catalytically active Ser-His-Asp triad. This was achieved by directed evolution to engineer the local environment of the Cys-His-Asp triad for improved catalytic activity. Deconvolution experiments have been conducted to investigate the effect of individual mutations. The authors also present crystal structures of two mutants and explore structure and dynamics of WT-CALB and several mutants in complex with 4-nitrophenyl benzoate as model substrate using docking / molecular dynamics studies. The reaction mechanism is investigated with QM/MM and QM cluster model calculations.

These results are interesting because a Ser->Cys conversion at the catalytic Ser-His-Asp triad in lipases generally reduces or shuts down catalytic activity. The experiments are well documented and the manuscript is well written. I do have doubts about the claim that the reaction mechanism differs in Ser-His-Asp vs Cys-His-Asp mutants of CALB based on QM/MM calculations (see detailed comments below).

Perhaps the discussion (and abstract) should point out that enzyme engineering and enzyme activity studies were conducted specifically (and only) for bulky substrates for which WT-CALB activity is low. Why was a substrate chosen for which WT-CALB activity is low? What would be the expected outcome for natural substrates of WT-CALB?

The authors point out that the highest activity was found for mutant QW4 with a Cys-His-Asp catalytic triad. The directed evolution was performed with Ser->Cys replaced. Could a directed evolution based on WT-CALB with conserved Ser-His-Asp triad perhaps have yielded a mutant with even higher catalytic activity?

The molecular dynamics simulations are carried out well and support the conclusions. The author list is missing from the Amber citation in the Supporting Information. It is important to give proper credit to support the software engineering efforts that enable science.

The QM/MM and QM cluster calculations employ the B3LYP density functional. The basis set 6-31G+(d,p) is not large but reasonable. However, B3LYP is clearly not state of the art and it is well known that any GGA and hybrid functional is missing dispersion interactions. I sincerely do not understand why studies are still performed with outdated density functionals that are known to be deficient while modern density functionals that remedy some of the worst deficiencies have been available in common software packages, including Gaussian that was used by the authors, for many years. The qualitative conclusions drawn by the authors are probably still correct, but I would encourage to re-optimize stationary points using a more reliable density functional such as wB97X-D or at the very least include Grimme type dispersion correction (B3LYP-D3) and check the effect on structure and relative energies.

Using QM/MM calculations, the authors come to the conclusion that the reaction mechanism differs in mutants QW10 (concerted) and QW4 (step-wise). This is a bold statement and I doubt that the calculations support the conclusions.

Regarding the exclusion of a step-wise mechanism for mutant QW10:

The manuscript states "An extensive QM/MM investigation showed that for the QW10 variant, the first step [...] works via a concerted mechanism [...]". What exactly do the authors mean with "an extensive QM/MM investigation"? The authors do not provide the reaction profile for a step-wise mechanism. I assume the author have investigated a step-wise mechanism and the transition state is higher in energy. This data must be discussed in the manuscript and corresponding data added to the supporting information, otherwise it is not possible to conclude that the concerted mechanism is preferred in QW10.

Regarding the exclusion of a concerted mechanism for QW4:

Clearly, the transition state for a concerted mechanism is harder to locate. The fact that the authors were not able to locate the transition state for a concerted mechanism in WT4 does not mean that it does not exist, nor that it might not be lower in energy. In particular, the choice of reaction coordinate is very important. Looking at Supplementary Figure S14, a 2D reaction profile was computed with the QM cluster model for WT4 with reaction coordinates C105(S)-Substrate(C) (distance between substrate carbonyl carbon and thiolate sulphur) and C105(S--H) (distance between thiol sulphur and hydrogen). The latter, C105(S--H) does not seem an appropriate reaction coordinate as it must be coupled to the H224(N--H) distance. I would expect H224(N--H) to be a more appropriate reaction coordinate. Most likely though, what is required, is a linear combination of C105(S--H) and H224(N--H) (e.g. commonly employed distance difference) that properly couples proton transfer coordinate and location of proton donor/acceptor atoms (N/S). The fact that there is a discontinuity in the 1D cut of the reaction profile presented in Figure S14 a) is a clear indication that the choice of reaction coordinate is poor. It also means that starting the scans from the products will give a different profile, which is unphysical.

Data availability:

The authors should provide the force field parameters for p-nitrophenyl benzoate and coordinates of

- i) docked structures
- ii) coordinates from equilibrated MD
- iii) QM/MM stationary points
- iv) QM cluster model stationary points

Sincerely,
Andreas Goetz

Reviewer #4 (Remarks to the Author):

In this manuscript, Cen and coauthors used laboratory evolution to identify variants of *Candida*

arctica lipase B (CALB) that are more active toward 4-nitrophenyl benzoate compared to the wild-type enzyme. They identified a highly active variant, QW4, which has a Cys-His-Asp catalytic triad not seen before in natural lipases, which are only known to have Ser-His-Asp as their catalytic residues. QW4 also harbors four additional mutations that help to optimize substrate binding. The QW4 mutant tolerates bulky substrates with similar or higher catalytic efficiency compared to a variant with the Cys switched back to Ser. Interestingly, when applied for kinetic resolution of chiral esters, QW4 has opposite enantioselectivity relative to wild-type CALB, but is able to produce high enantiomeric excesses nonetheless. Structural studies and MD simulations revealed alterations in structural dynamics that facilitate substrate binding in the QW4 mutant. QM/MM calculations suggested that QW4 and the corresponding mutant with Ser switched back to Cys function via distinct mechanisms. The enzyme with a Ser-His-Asp triad appears to have a concerted mechanism, while the Cys-His-Asp mutant QW4 has a two-step mechanism in which deprotonation of the nucleophile occurs before nucleophilic attack on the ester. Overall, this study is well executed and demonstrates that a Cys-His-Asp catalytic triad can support high enzymatic activity in the CALB scaffold. This raises the question of whether a Ser to Cys swap has ever occurred during lipase evolution, and opens up new possibilities for engineering lipases with substrate selectivity different from natural enzymes. This work is therefore of significant interests to biochemists and protein engineers and is suitable for publication in Nature Communications with minor revisions.

Comments:

1. It would be helpful if the authors further explained the grouping of residues for mutation that is shown in Figure 2a, especially positions 276 and 285, since they are not the closest together in the primary sequence or secondary structure of the enzyme.
2. On page 3, in the second paragraph, the authors present the surprising result that the most active mutant in their screen has a Ser to Cys mutation. However, the kinetic analysis presented in Table 1 shows that the catalytic efficiency is actually about two-fold lower than that of the wild-type enzyme. Can the authors speculate on why this mutant came out of the screen despite its lower activity? (Expression level? Was the screen performed under conditions that select for improved k_{cat} ?)
3. When the authors discuss their screening results, specific activities are given in $\mu\text{M min}^{-1} \text{OD}^{-1}$. It would be helpful if they also mentioned the fold change compared to wild-type in the text for ease of comparison.
4. On page 3, at the end of the first paragraph, the authors mention that iterative saturation mutagenesis at sites D, E, and F did not result in any improved mutants. Did mutations at these sites make activity worse than the activity of the parent enzyme, or just no better?

Overall, this manuscript is technically sound and informs our understanding of lipase evolution, mechanism, and engineering. It is therefore suitable for publication in Nature Communications after these comments are addressed.

Response to reviewers' comments

Reviewer #1:

Comment (1):

Authors describe the preparation of different new artificial calB variants by directed evolution, focus on the modification on particular positions on the structure.

They found some variants with better improvement in activity and some results in inversion in enzyme enantioselectivity. However, the main point in the idea which is the cysteine-lipases is not clear to be such relevant. They found activity in the lipase after mutation catalytic serine by cysteine, conserving around 50% the initial activity, however the final better results in the new artificial enzyme is based on other modifications, W104, A281, A282 and V149.

Indeed, the QW10 variant, which contain these 4 mutations conserving intact the catalytic triad is much better in general than the cysteine 105 containing variant: Hydrolytic activity of several substrates identical, and much better for QW10 with 5 and quite similar with substrate 1. Furthermore, a critical point is the selectivity where in the reaction in table 2, both QW10 and QW4 showed the same inversion in enantioselectivity, moderate value, but with substrate 11, an E value of 63S for QW10 against 13S for QW4 (also the reaction was faster for QW10). Also for rac-15 QW10 was better.

It has been well described that W104 has a critical role in the enantioselectivity of this enzyme. Even authors previously published critical changes in selectivity for this enzyme in JACS, 2013, and other authors (Hult, *chembiochem*, 2010, *OBC*, 2011). Also they employed similar amino acid positions such as 282 in combination with W104 for variants showing better activity and selectivities.

Therefore, the results seem demonstrate that the relevant alterations in the better variants are W104, A282 and A281 (oxyanion) instead of the modification of the catalytic serine, which is harmful in many cases.

Response (1):

Thank you very much for your important comments.

We completely agree with your comments concerning the influence of mutations on the positions of W104, A282 and A281 (oxyanion) on the activity and enantioselectivity of variants QW4 and QW10, especially the critical role of W104V mutation in the enantioselectivity inversion of these variants.

Concerning the critical parts, it is necessary to consider the following background information: For lipases, only a few reports of cysteine analogues have been reported, all resulting in greatly reduced enzyme activity (Supplementary Table S1, Page: S32). To date, protein engineering of nucleophile-exchanged mutants with improved activity has never been achieved. *In our work, we created by laboratory evolution a highly active "cysteine-lipase" with a hybrid Cys-His-Asp catalytic triad, QW4 variant, which shows a 40-fold higher catalytic efficiency than wildtype CALB in the hydrolysis of the model substrate (4-nitrophenyl benzoate)*. Indeed, many experiments focusing on the simple exchange of nucleophilic serine and cysteine cause notable reduction in activity. Our results show *for the first time* that lipases can indeed tolerate the nucleophile exchange from serine to cysteine in the Cys-His-Asp catalytic triad, while the local

environment around the distinct catalytic triads should be accurately manipulated by laboratory evolution.

Indeed, the question of cysteine introduction in place of serine at catalytically active triads (and understanding the result on a molecular level) constitutes a classical challenge in enzymology. In addition to solving the synthetic chemical problem of substrate acceptance, our present paper provides a final answer to the origin of the mutational effect.

Actually, except for substrate 5, QW4 has similar or better activity than QW10 for all substrates with similar structure as the model substrate (**1**) in the screening step. Of course, Reviewer 1# is right that with substrate **11**, QW10 is better than QW4 in the respect of activity and selectivity. However, QW4 and QW10 have similar performance for all other substrates such as **13**, **15**, **17**, **19**, **21** (Table 2, Table S3). It is reasonable that QW4 variant does not show much better performance than QW10 for these expanded substrate because they are not the model compound used in screening. *These cases indicate that the interconversion of nucleophilic serine and cysteine in this work is indeed successful, the activity of “cysteine-lipase” variant (QW4) is not only much better than WT, but also completely comparable with the optimized variant (QW10) with the native Cys-His-Asp catalytic triad.*

Comment (2):

Most studies in this work have been performed with model substrate and also it would be important testing the variants, specially WQ10, in other interesting reactions where high alteration in selectivity, enantio or preference can be demonstrated.

Response (2):

Thank you very much for your good suggestions.

We have added more reactions under the catalysis of QW4 and QW10, such as the hydrolytic kinetic resolution reactions of 1-phenylpropyl acetate (*rac*-**13**), 1-phenylpentyl acetate (*rac*-**17**), dimethyl cyclohexane-1,2-dicarboxylate (*trans*-**21**). For *rac*-**13** and *rac*-**17**, both QW4 and QW10 showed better activity and stereoselectivity than QW1. It is noteworthy that *rac*-**17** with a large alkyl group also can be accepted by QW4 and QW10 with moderate selectivity, in contrast to WT CALB, which has no activity toward this substrate due to the limited space of the alcohol-binding pocket. In the case of KR of *trans*-**21**, surprisingly, QW1 displays unexpectedly excellent performance for this specific substrate like WT CALB. Please see these reactions in Supplementary Table S3 (Page: S34-35).

We also tested the nonaqueous transesterification of *p*-nitrophenyl benzoate (**1**) using methanol (see Supplementary Figure S4, Page: S12). Interestingly, QW4 also showed better performance than WT CALB and QW10 variant. All of these new results demonstrate excellent synthetic utility of QW10 variant and the “cysteine-lipase” variant (QW4).

Reviewer #2:

Comment (1):

Cen et al. – Artificial cysteine-lipases with high activity and altered catalytic mechanism created by laboratory evolution.

This paper reports on the engineering of CALB into a “cysteine lipase”. This is the first time that a lipase is successfully mutated from a serine lipase into a cysteine lipase, and it works a proof of concept, influencing thinking in the field. The work is technically sound and well written.

The only concerns come from the points below, which should be addressed before publication.

Fig. S1 and introduction. In the manuscript introduction, authors refer to the proteases alignment in fig. S1 as phylogenetic analysis of proteases. A phylogenetic analysis requires a more consistent amount of data to be compared and, moreover, a detailed phylogenetic tree based on a conserved protein domain, on the amino acidic stretch containing the active site or on the full protein/enzyme sequence. The alignment of four sequences is indicative to show the Cys to Ser exchange between different organisms but not enough to support the concept of phylogeny.

Response (1):

Thank you very much for your important comments and good suggestion.

We have redone the multiple sequence alignment and phylogenetic analysis of serine proteases and cysteine proteinase. Please find them in Supplementary Figure S1 (Page: S8-9).

Comment (2):

*Page 5. Crystal Structure characterization. The refinement resolution indicated for QW10 is 2.1 Å. In the supplementary table S4 is indicated however a resolution of 1.72 Å for data collection with reasonable data statistics. What is the reason to cut back the refinement resolution to 2.1 Å?

Response (2):

Thank you very much for your comments. We now report the following new improved results:

We tried different resolution cut-off in structural refinement. When full resolution (1.72 Å) was used in refinement, R_{factor} was 0.2297 and R_{free} was 0.2687. They were incredibly high. After long-time checking and testing, we found that resolution cut-off is the key factor to lower the $R_{\text{factor}}/R_{\text{free}}$. As shown in the below table, resolution cut-off to 2.1 Å gave the best refinement statistic.

Highest resolution used in refinement (Å)	1.72	1.8	1.9	2.0	2.1
R_{free}	0.2687	0.2635	0.2594	0.2600	0.2560
R_{factor}	0.2297	0.2205	0.2203	0.2252	0.2180

Comment (3):

*Supporting Information. Crystallization and X-ray Structural Analysis. Second last line. Supplementary Table S3 should be S4 instead.

Response (3):

Thank you very much for your careful checking! We have corrected the mistake in the revised Supporting Information (Page: S6).

Comment (4):

*Supporting Information. Supplementary Figure S4. Please indicate the sigma level used to contour the map.

Response (4):

Thank you very much for your suggestion. The sigma level used for 2Fo-Fc density map in Supplementary Figure S4 (renumbered as Figure S5) was 1σ . We have added this information in the Figure legend of updated Supplementary Figure S5 (Page: S13).

Comment (5):

*Supporting Information. Supplementary Table S4. The data collection statistic should be indicated for the resolution used for refinement ie. 2.1 Å of QW10. Please present data for the last resolution bin too.

Response (5):

Thanks! We have presented the updated dataset (1.88 Å, re-processed by XDS) of QW10 with the last resolution in the revised Table S4 in Supporting Information (Page S36).

Comment (6):

*Supporting Information. Supplementary Table S4. Data collection. Please indicate Wilson B-factor.

Response (6):

Thank you very much for your suggestion. We have added Wilson B-factor in Supplementary Table S4 (Page S36).

Comment (7):

*Supporting Information. Supplementary Table S4. Refinement. % Ramachandran outliers, % Ramachandran favored, % rotamer outliers, should be indicated.

Response (7):

Sorry we forgot to include the detailed information of Ramachandran statistics in Supplementary Table S4. We have taken your suggestions and added them in the revised Supplementary Table S4 (Page S36).

Comment (8):

*Supporting Information. Supplementary Table S4. The R_{free}/R_{factor} are considerably high for QW10 when compared to QW4 in both forms. Also the number of ligands (401) and waters (442) is high for QW10 when compared to QW4 in both forms, given also the different resolutions. It appears that there are some problems with the refinement of this structure. What is like the plot of R_{free}/R_{factor} vs resolution for QW10? How was the space group determined?

Response (8):

The diffraction quality of QW10 crystals could only be obtained in the presence of N, N-bis-(3-D-Gluconamidopropyl) deoxycholamide (Big CHAP, deoxy), a detergent molecule containing 60 non-hydrogen atoms. We found that there are 5 detergent molecules, i.e. 300 detergent molecule atoms, surrounding two QW10 protein chains in one asymmetric unit. This result explains why the number of ligands is high for QW10 when compared to QW4.

The plot of R_{free}/R_{factor} vs resolution for QW10 is listed in the below figure: a, previous QW10 with resolution of 1.72 Å; b, previous QW10 with resolution cut off to 2.1 Å; c, re-processed QW10 dataset with resolution of 1.88 Å.

The data set of QW10 was originally processed by using the XIA2 package at SSRF beamline 17U1, with space group $P2_1$ determined. We agree with the reviewer that there might be some problems with the refinement of this structure. We first tried different resolution cut-off in structural refinement. When full resolution (1.72 Å) was used in refinement, R_{factor} was 0.2297 and R_{free} was 0.2687. They were incredibly high. After long-time checking and testing, we found that resolution cut-off is the key factor to lower the $R_{\text{factor}}/R_{\text{free}}$. As shown in the below table, resolution cut-off to 2.1 Å gave the best refinement statistic.

Highest resolution used in refinement (Å)	1.72	1.8	1.9	2.0	2.1
R_{free}	0.2687	0.2635	0.2594	0.2600	0.2560
R_{factor}	0.2297	0.2205	0.2203	0.2252	0.2180

We next tried to re-process the data set using the XDS package and used the new scaled dataset for structure refinement. There are three possible space groups, $P2_12_12_1$, $P2_1$, and $P1$. The probability of $P2_12_12_1$ based on the observed systematic absences was only 0.490. Pseudo symmetry was detected when using $P2_12_12_1$ space group and R_{factor} resulted above 0.25. When we lowered the symmetry to $P2_1$, probability based on the observed systematic absences became 0.957. Space group confidence was 0.935. The final $R_{\text{free}}/R_{\text{factor}}$ was reduced to 0.2383/0.2135 when resolution cut-off was set to 1.88 Å. This statistic is much better than the one from the scaled dataset using the XIA2 package. This time, some long tails of the detergent were removed for their poor map density. Therefore, there are 342 detergent molecule atoms and only two non-detergent ligands in the updated structure. We also tried to use the $P1$ space group for structure refinement, however, the final $R_{\text{free}}/R_{\text{factor}}$ 0.2512/0.2301, much higher than the one from $P2_1$ data-set. Based on this data, the space group for QW10 dataset should be $P2_1$, a space group in good agreement with what we described in the original manuscript. We added them in the revised Supporting Information (Page S6) for the benefit of the crystallography community.

We also compared the updated structure of QW10 with the old one by aligning two overall structures and the two active sites. The RMSD values were 0.117 and 0.118 respectively, suggesting two structures are almost the same. The new calculation results based on this updated QW10 structure suggest that the final conclusions are essentially the same as originally proposed.

The aligning figure of the two active sites is shown below, with relevant residues labeled: light blue for previous structure; green for updated structure.

Data statistics are attached below and in Supplementary Table S4 (Page S36).

	QW10 previous	now
Data collection		
Space group	$P2_1$	$P2_1$
Cell dimensions		
a, b, c (Å)	47.0, 92.5, 78.4	47.0, 92.5, 156.0
α , β , γ (°)	90.0, 95.5, 90.0	90.0 90.013, 90.0
Resolution(Å)	41.74 - 1.72 (1.76 - 1.72)	46.26 - 1.88 (1.91 - 1.88)
R_{merge} (%)	12.2 (78.5)	13.2 (99.3)
Completeness(%)	97.0 (94.0)	98.8 (99.2)
Average(I/ σ)	7.8 (1.9)	9.4 (2.1)
Redundancy	7.0	6.6
Wilson B-factor(Å ²)	19.23	23.67
Refinement		
Resolution(Å)	41.74 - 2.10	42.26 - 1.88
No. reflections used	37542	106351
$R_{\text{work}} / R_{\text{free}}$	0.2180 / 0.2560	0.2135 / 0.2383
Asymmetric unit	2	4
No. non-H atoms		
Protein	4714	9336
Ligand	401	344
Water	442	1228
Overall B factor(Å ²)	25.3	25.6
Ramachandran		
Favored (%)	96.24	96.36
Outliers (%)	0.00	0.16
rotamer outliers (%)	1.94	0.29
RMSD		
Bond lengths(Å)	0.009	0.003
Bond angles (°)	1.275	0.924

Comment (9):

*Results. Authors state that the inconsistency between specific activities and catalytic kinetics between QW1 and WT CALB is due to the different expression levels of the mutants. In order to obtain comparable kinetics data, equal amounts of purified variants should be the starting point for the subsequent generation of data. The equal amount of protein is not correlated to the expression level of the mutant but it is a downstream normalization. Moreover, Authors compare the catalytic efficiency of the variants QW2, QW3, QW4. If true what previously mentioned, also in this case the expression level should be considered and the starting amount of proteins equalized to have comparable data.

Response (9):

Thank you very much for your important comments and suggestion.

We have determined the protein expression level of WT, QW1-QW4, and listed the data in the following table. It can be found the WT has the best expression level, QW1-QW2 and QW4 are similar.

Concerning the relationship between the kinetics data and the expression level, the reviewer's comment is completely right, namely the kinetics data has no direct relation with the expression level because equal amounts of purified protein are used as the starting point for the kinetics determination of QW1-QW4 and WT. The equal amount of protein is not correlated to the expression level of the mutant but it is a downstream normalization. QW2 has a small kinetic data only implying the low activity of one protein molecule. However, the specific activity of different CAL B mutants is much more complicated, although it is a more convenient assessment method without the requirement of protein purification. The specific activity ($\mu\text{M}\times\text{Min}^{-1}\times\text{OD}^{-1}$) is defined as the amount of substrate the enzyme converts, per amount of wet cells in the enzyme culture, per unit of time. The specific activity of different variants correlate not only with the kinetic data (catalytic efficiency or protein activity) of the mutants, and the expression level of the mutants, but also with the growth ability or cell density (OD600) of the mutants. Indeed, the growth ability or cell density (OD600) of the mutants are somehow different, and could be influenced by the inoculation, the shaking rate and other cultivation conditions. Thus we think the original expression of "The apparent inconsistency between specific activities and catalytic kinetics is due to the different protein expression level of various mutants" is not suitable, and should be changed. We have revised this sentence (Line 28-30, Right column, Page 3).

Enzymes	Protein expression level (mg/L)
WT	1.23
QW1	0.54
QW2	0.57
QW3	0.89
QW4	0.43

Comment (10):

*Reference #10. "subtillisin" should be "subtilisin"

*Reference #14. "viology" should be "virology"

Response (10):

Thank you very much for your suggestions. We have revised these errors in the References No.

Reviewer #3:

Comment (1):

The authors have engineered highly active lipases for bulky substrates, employing *Candida antarctica* lipase B (CALB), notably some of which have Ser replaced by Cys in the catalytically active Ser-His-Asp triad. This was achieved by directed evolution to engineer the local environment of the Cys-His-Asp triad for improved catalytic activity. Deconvolution experiments have been conducted to investigate the effect of individual mutations. The authors also present crystal structures of two mutants and explore structure and dynamics of WT-CALB and several mutants in complex with 4-nitrophenyl benzoate as model substrate using docking / molecular dynamics studies. The reaction mechanism is investigated with QM/MM and QM cluster model calculations.

These results are interesting because a Ser->Cys conversion at the catalytic Ser-His-Asp triad in lipases generally reduces or shuts down catalytic activity. The experiments are well documented and the manuscript is well written. I do have doubts about the claim that the reaction mechanism differs in Ser-His-Asp vs Cys-His-Asp mutants of CALB based on QM/MM calculations (see detailed comments below).

Response (1):

Thank you very much for your important comments.

Comment (2):

Perhaps the discussion (and abstract) should point out that enzyme engineering and enzyme activity studies were conducted specifically (and only) for bulky substrates for which WT-CALB activity is low. Why was a substrate chosen for which WT-CALB activity is low? What would be the expected outcome for natural substrates of WT-CALB?

Response (2):

Thank you very much for your important suggestions. We have added one sentence in the Abstract and Discussion, respectively, pointing out that these bulky substrates are essentially not accepted by WT-CALB. Please see them in line 5-6 in Abstract (Page 1), and line 23-24 in the section of Discussion and conclusions (Right column, Page 8).

In order to examine the subtle influence of the nucleophile exchange Ser → Cys of CALB on its activity, the new substrate 4-nitrophenyl benzoate was chosen, which is hardly turned over by WT-CALB. According to the previous studies, the direct Ser→Cys exchange in lipases is known to shut down activity. Here again, acceptable results were observed.

Comment (3):

The authors point out that the highest activity was found for mutant QW4 with a Cys-His-Asp catalytic triad. The directed evolution was performed with Ser->Cys replaced. Could a directed evolution based on WT-CALB with conserved Ser-His-Asp triad perhaps have yielded a mutant with even higher catalytic activity?

Response (3):

Thank you for your suggestion.

You are probably right. A directed evolution study based on WT-CALB with conserved Ser-His-Asp triad could provide a serine-lipase mutant with higher catalytic activity. However, that would go beyond the realm of the present work, and would not throw light on the origin of serine to cysteine exchange effects.

Comment (4):

The molecular dynamics simulations are carried out well and support the conclusions. The author list is missing from the Amber citation in the Supporting Information. It is important to give proper credit to support the software engineering efforts that enable science.

Response (4):

We appreciate the reviewer's careful checking and good suggestion.

We have added the full author list in the citation of Amber in the Supporting Information (Ref 24, Page S39). Moreover, the full author list in the citation of Gaussian 09 in the Supporting Information has also been added (Ref 19, Page S38).

Comment (5):

The QM/MM and QM cluster calculations employ the B3LYP density functional. The basis set 6-31G+(d,p) is not large but reasonable. However, B3LYP is clearly not state of the art and it is well known that any GGA and hybrid functional is missing dispersion interactions. I sincerely do not understand why studies are still performed with outdated density functionals that are known to be deficient while modern density functionals that remedy some of the worst deficiencies have been available in common software packages, including Gaussian that was used by the authors, for many years. The qualitative conclusions drawn by the authors are probably still correct, but I would encourage to re-optimize stationary points using a more reliable density functional such as wB97X-D or at the very least include Grimme type dispersion correction (B3LYP-D3) and check the effect on structure and relative energies.

Response (5):

We appreciate the referee's comment, and have added some new computational data. Although density functional with dispersion have been recommended for many systems, B3LYP has still been widely used in many recent studies and has led to quite similar optimized geometry as those obtained by using wB97X-D or B3LYP-D3. A commonly adopted protocol is to use B3LYP for optimization, followed by energy correction using the dispersion density function such as B3LYP-D3 based on single-point energy calculations. For example, in a recent QM/MM study on CALB, Thiel et al used B3LYP for the reaction mechanism study and used wB97X-D (B3LYP-D2, B3LYP-D3) for single point energy calculations for the stationary points (*ACS Catal.* 2017, 7, 115–127). It was found the single point energies generally only vary by several kcal/mol, a minor difference. Similar results were also observed in other recent research by Liu et al. (*Phys. Chem. Chem. Phys.*, 2018, 20, 17342-17352).

Taking the referee's suggestion, we have re-optimized the stationary points on the reaction path for both QW10 and QW4. As expected, the comparison shows that the optimized geometries using both density functional are almost the same (Figure R1 and R3), and the difference in the energies is within a couple of kcal/mol (Figure R2 and R4). In summary, inclusion of dispersion still leads to the same qualitative conclusion so that it is not necessary for the system being investigated. We have added the re-optimization results of the stationary points by QM/MM

(AMBER/wB97X-D/6-31G+(d,p)) for both QW10 and QW4 in Supplementary Figure S10-S12 (Page S18-S20), and also updated Figure 5 (Page 7), the reaction profiles of QW10 and QW4 calculated by using QM/MM (AMBER/ ω B97X-D/6-31G+(d,p)).

Figure R1. The stationary points of the QW10 variant calculated by QM/MM (AMBER/B3LYP/6-31G+(d,p) vs. AMBER/wB97X-D/6-31G+(d,p)).

Figure R2. The reaction profile for the QW10 variant calculated by QM/MM (AMBER/B3LYP/6-31G+(d,p) vs. AMBER/wB97X-D/6-31G+(d,p)). Here the reaction coordinate is the nucleophilic attack of the hydroxyl group in S105 (OH) on the carbonyl carbon of the substrate.

QW4 (QM/MM)

Figure R3. The stationary points of the QW4 variant calculated by QM/MM (AMBER/B3LYP/6-31G+(d,p) vs. AMBER/wB97X-D/6-31G+(d,p)).

QW4 (QM/MM)

Figure R4. The reaction profile for the QW4 variant calculated by QM/MM (AMBER/B3LYP/6-31G+(d,p) vs. AMBER/wB97X-D/6-31G+(d,p)). Here the reaction coordinate is the nucleophilic attack of the thiol group in C105 (SH) on the carbonyl carbon of the substrate.

Comment (6):

Using QM/MM calculations, the authors come to the conclusion that the reaction mechanism differs in mutants QW10 (concerted) and QW4 (step-wise). This is a bold statement and I doubt that the calculations support the conclusions.

Regarding the exclusion of a step-wise mechanism for mutant QW10:

The manuscript states "An extensive QM/MM investigation showed that for the QW10 variant, the first step [...] works via a concerted mechanism [...]". What exactly do the authors mean with "an extensive QM/MM investigation"? The authors do not provide the reaction profile for a step-wise mechanism. I assume the author have investigated a step-wise mechanism and the transition state is higher in energy. This data must be discussed in the manuscript and corresponding data added to the supporting information, otherwise it is not possible to conclude that the concerted mechanism is preferred in QW10.

Response (6):

Yes, in addition to the concerted mechanism reported in the manuscript, we had also considered the possibility of a step-wise mechanism for QW10. For the reaction to occur via a two-step mechanism, the stable structure of the ionic pair form of the zwitterionic ion could not be obtained from scanning the reaction coordinates corresponding to the proton transfer from the Ser105 hydroxyl to the His224 Ne. Any attempt to optimize the geometry closest to the ionic pair form from the potential energy surface always led to the neutral form. In addition, the energy kept increasing with the decrease of the reaction coordinate and no transition corresponding to the proton transfer from the Ser105 hydroxyl to the His224 Ne could be located, ruling out the possibility of step-wise mechanism. In the revised manuscript, we have included the discussion (see line 29-41 in left column, Page 8) and corresponding data (Figure S11, page S19) for investigating the two-step mechanism of QW10.

Comment (7):

Regarding the exclusion of a concerted mechanism for QW4:

Clearly, the transition state for a concerted mechanism is harder to locate. The fact that the

authors were not able to locate the transition state for a concerted mechanism in WT4 does not mean that it does not exist, nor that it might not be lower in energy. In particular, the choice of reaction coordinate is very important. Looking at Supplementary Figure S14, a 2D reaction profile was computed with the QM cluster model for WT4 with reaction coordinates C105(S)-Substrate(C) (distance between substrate carbonyl carbon and thiolate sulphur) and C105(S--H) (distance between thiol sulphur and hydrogen). The latter, C105(S--H) does not seem an appropriate reaction coordinate as it must be coupled to the H224(N--H) distance. I would expect H224(N--H) to be a more appropriate reaction coordinate. Most likely though, what is required, is a linear combination of C105(S--H) and H224(N--H) (e.g. commonly employed distance difference) that properly couples proton transfer coordinate and location of proton donor/acceptor atoms (N/S). The fact that there is a discontinuity in the 1D cut of the reaction profile presented in Figure S14 a) is a clear indication that the choice of reaction coordinate is poor. It also means that starting the scans from the products will give a different profile, which is unphysical.

Response (7):

Thank you very much for your comments.

In addition to the two-step mechanism we reported, we also examined the possibility of concerted mechanism for QW4, to see if the proton transfer from Cys105 thiol Hydrogen to His224 Nitrogen would occur simultaneously with the nucleophilic attack of Cys105 thiol Sulphur to the substrate carbonyl carbon atom. Therefore, the two reaction coordinates in 2D scan should be C105(S)-Substrate(C) (the distance between substrate carbonyl carbon and thiolate sulphur) and C105(S--H) (the distance between thiol sulphur and hydrogen).

To describe the proton transfer, either Cys105 (S--H) (the distance between thiol sulphur and hydrogen) or Cys105 (SH---His224 N ϵ) (the distance between thiol hydrogen and His224 N ϵ) can be used. We scanned the proton transfer for QW4 using the two coordinates and actually both scans gave similar profiles (Figure R5). Further, no stable ionic pair nor transition state could be located for both scans by optimizing the respective near-stationary points on the scan profiles.

Figure R5. QM/MM Potential surface scan corresponding to the proton transfer in QW4, QM is calculated using B3LYP/6-31+G (d,p). **a**, The reaction coordinate is the distance between the Sulphur and Hydrogen of the Cys105 thiol group. **b**, The reaction coordinate is the distance between the thiol Hydrogen of Cys105 and the epsilon Nitrogen of His224.

We also re-ran the complete 2D scan using C105(S)-Substrate(C) (distance between substrate carbonyl Carbon and thiolate Sulphur) and Cys105 (SH)---His224 (N ϵ) (distance between thiol Hydrogen and His224 N ϵ) using QM cluster method. From the new scan, again no transition state

could be located corresponding to the concerted mechanism, indicating concerted mechanism is not viable for QW4. Please see the new Supplementary Figure S13 (Page S21) and Figure S15 (Page S23).

Comment (8):

Data availability:

The authors should provide the force field parameters for *p*-nitrophenyl benzoate and coordinates of i) docked structures

ii) coordinates from equilibrated MD

iii) QM/MM stationary points

iv) QM cluster model stationary points

Response (8):

Thank you for your good suggestions.

The force field parameters and the coordinates associated with the docked structures, the equilibrated MD, and the optimized geometries of the stationary points from QM/MM calculations as well as from the QM cluster calculations are now supplied as Source Data files.

Reviewer #4:

Comment (1):

In this manuscript, Cen and coauthors used laboratory evolution to identify variants of *Candida arctica* lipase B (CALB) that are more active toward 4-nitrophenyl benzoate compared to the wild-type enzyme. They identified a highly active variant, QW4, which has a Cys-His-Asp catalytic triad not seen before in natural lipases, which are only known to have Ser-His-Asp as their catalytic residues. QW4 also harbors four additional mutations that help to optimize substrate binding. The QW4 mutant tolerates bulky substrates with similar or higher catalytic efficiency compared to a variant with the Cys switched back to Ser. Interestingly, when applied for kinetic resolution of chiral esters, QW4 has opposite enantioselectivity relative to wild-type CALB, but is able to produce high enantiomeric excesses nonetheless. Structural studies and MD simulations revealed alterations in structural dynamics that facilitate substrate binding in the QW4 mutant. QM/MM calculations suggested that QW4 and the corresponding mutant with Ser switched back to Cys function via distinct mechanisms. The enzyme with a Ser-His-Asp triad appears to have a concerted mechanism, while the Cys-His-Asp mutant QW4 has a two-step mechanism in which deprotonation of the nucleophile occurs before nucleophilic attack on the ester. Overall, this study is well executed and demonstrates that a Cys-His-Asp catalytic triad can support high enzymatic activity in the CALB scaffold. This raises the question of whether a Ser to Cys swap has ever occurred during lipase evolution, and opens up new possibilities for engineering lipases with substrate selectivity different from natural enzymes. This work is therefore of significant interests to biochemists and protein engineers and is suitable for publication in Nature Communications with minor revisions.

Response (1):

Thank you very much for your important comments.

Comment (2):

1. It would be helpful if the authors further explained the grouping of residues for mutation that is shown in Figure 2a, especially positions 278 and 285, since they are not the closest together in the primary sequence or secondary structure of the enzyme.

Response (2):

Thank you very much for your suggestion.

It is well known that the helix 277-288 has important influence on the performance of CALB. And the best mutants in this work also confirmed this effect, especially A281 and A282 mutations. Thus positions A281, A282, L278 and I285 in this helix 277-288 were chosen for mutations. Moreover, many studies of directed evolution have indicated that the effect of multiple mutations in one multiple-point variant is cooperative rather than additive. In order to display the possible cooperative effect between mutations at A281 and A282 as clearly as possible, these two positions were combined. And the other two positions (L278 and I285) were also grouped likewise due to rational reasons.

Comment (3):

2. On page 3, in the second paragraph, the authors present the surprising result that the most active mutant in their screen has a Ser to Cys mutation. However, the kinetic analysis presented in Table 1 shows that the catalytic efficiency is actually about two-fold lower than that of the wild-type enzyme. Can the authors speculate on why this mutant came out of the screen despite its lower activity? (Expression level? Was the screen performed under conditions that select for improved kcat?)

Response (3):

Thank you very much for your important comments and suggestion. See also our comments regarding the question of the other reviewer above.

We have determined the protein expression level of WT, QW1-QW4, and listed the data in the following table. It can be found the WT has the best expression level, QW1-QW2 and QW4 are similar.

The kinetics data has no direct relation with the expression level because equal amounts of purified protein are used as the starting point for the kinetics determination of QW1-QW4 and WT. The equal amount of protein is not correlated to the expression level of the mutant but it is a downstream normalization. QW2 has a small kinetic data only implying the low activity of one protein molecule. However, the specific activity of the different CALB mutants is much more complicated, although it is a more convenient assessment method without the requirement of protein purification. The specific activity ($\mu\text{M}\times\text{Min}^{-1}\times\text{OD}^{-1}$) is defined as the amount of substrate the enzyme converts, per amount of wet cells in the enzyme culture, per unit of time. The specific activity of different variants correlate not only to the kinetic data (catalytic efficiency or protein activity) of the mutants, and the expression level of the mutants, but also to the growth ability or cell density (OD600) of the mutants. Indeed, the growth ability or cell density (OD600) of the mutants are somehow different, and will be influenced by the inoculation, the shaking rate and other cultivation conditions. Thus we think the original expression of “The apparent inconsistency between specific activities and catalytic kinetics is due to the different protein expression level of various mutants” is not suitable, and should be changed. We have revised this sentence (Line 28-30, Right column, Page 3).

Enzymes	Protein expression level (mg/L)
WT	1.23
QW1	0.54
QW2	0.57
QW3	0.89
QW4	0.43

Comment (4):

3. When the authors discuss their screening results, specific activities are given in $\mu\text{M min}^{-1}$ OD-1. It would be helpful if they also mentioned the fold change compared to wild-type in the text for ease of comparison.

Response (4):

Thank you very much for your suggestion.

We have added the fold changes of specific activities compared to WT CALB in the main text, please see them in Lines 10-11, 15 and 22, Right column, Page 3.

Comment (5):

4. On page 3, at the end of the first paragraph, the authors mention that iterative saturation mutagenesis at sites D, E, and F did not result in any improved mutants. Did mutations at these sites make activity worse than the activity of the parent enzyme, or just no better?

Response (5):

Thank you very much.

No mutants better than QW4 variant were found in the new libraries created by further mutations at sites D, E and F.

Comment (6):

Overall, this manuscript is technically sound and informs our understanding of lipase evolution, mechanism, and engineering. It is therefore suitable for publication in Nature Communications after these comments are addressed.

Response (6):

Thank you very much for your important comment.

Statement about other revisions:

1. Five source data files are provided, and the description of these source data files is listed as follows.

File Name: Supplementary Data 1

Description: Detailed data of Table 1, Fig. S3, Fig. S4, Fig. S11, Fig. S13, Fig. S15 and Table

S2.

File Name: Supplementary Data 2

Description: The force field parameters for p-nitrophenyl benzoate.

File Name: Supplementary Data 3

Description: The coordinates of QM/MM stationary points of QW10 and QW4 mutants.

File Name: Supplementary Data 4

Description: One zip file containing the PDB files of the docked structures and the equilibrated MD structures of WT, QW2, QW4 and QW10 variants.

File Name: Supplementary Data 5

Description: One zip file containing X-ray Structure Validation Reports for 6ISQ (QW4-oxidized), 6ISR (QW4-unoxidized) and 6ISP (QW10).

2. The reference to these source data files has been added in the data availability statement (Right column, Page 9).

Reviewers' comments:

Reviewer #1 (Remarks to the Author):

The manuscript has been improved in this new revised version by adding new examples which clearly demonstrate the potential applicability of the creating these artificial cysteine-lipases. The producing altered lipases including Cys in the active site is known that cause a decrease on the activity, but an important issue is the role of the SH in the catalytic activity and selectivity of the enzyme instead of the OH.

Here the authors provide new examples to enhance the advantage of creating a redesign new active site by introducing the Cys in the active site and other mutations.

Interesting results with important increased on activity on the new created lipase in reactions in organic solvents. This open another question, which is the stability tolerance of this new artificial enzyme compared to the wild type??

New examples on improvement or maintaining the selectivity has been included, for example with the substrate 21, but in the table no ee (or E value) has been shown although HPLC shows a only one enantiomer. This result must be commented on the text in the main manuscript, it was not mention. The conversion is low, is was possible to get higher conversion to evaluate the selectivity??

In HPLC profile for the hydrolysis of 15, differences on the retention time of the enantiomers between the sample and the results with the catalysts can be observed (FigS18) From 12-14 min for sample 20-26 for the rest, why??? The experimental conditions were different??? An explanation must be included.

Reviewer #2 (Remarks to the Author):

All my points have been fully addressed. I strongly support the publication of this work. Sincerely,
Dr. M. Cianci

Reviewer #3 (Remarks to the Author):

The authors have addressed most of my concerns. This article is publishable after checking and addressing my comment on the reaction mechanism below.

In particular, new data indicates that, contrary to QW10, for QW4 a 2-step mechanism is favorable over a concerted mechanism. However, I would suggest to reformulate the sentence on page 8, right column, top ("However, no transition state could be located, indicating the improbability of a concerted mechanism for variant QW4"). This statement seems contradictory to the 2D scan presented in Figure S13, which seems to indicate that there is a transition state. More appropriate would be state that the scan indicated that a concerted mechanism is less favorable.

I am attaching an annotated version of the 2D potential energy surface scan from Figure S13. I indicated what I understand to be the position of the reactant, TS1, IM1, TS2 and product for the 2-step mechanism. I would like to ask the authors to label these points in their Figure, as it is hard to understand otherwise what is going on.

Unfortunately all stationary points are outside of the scan region. The reactant is at a C105(S)-C(substrate) distance of around 3.7 Angstrom (inferred from QM/MM data in Figure R3) and should be lower in energy than the zero point of the 2D scan. Similarly, TS1 should be at a lower energy. Similarly, IM1 will be at a C105(SH)-H224(Ne) distance shorter than 1.4 Angstrom, the same for the product, which will also be at a C(105)S-C(substrate) distance of around 1.8 Angstrom (the 2D scan stops at 2.0).

It is thus impossible to infer quantitative data about the energetics from the plot. However, unless I misunderstand the reaction coordinates and location of stationary points, it looks to me like there

is a transition state for a concerted reaction, as indicated in the attached figure. It is probably higher in energy than TS1 for the 2-step mechanism, but it would be wrong to say that such a transition state does not exist.

The authors should check this (I am not asking for an optimization of the transition state but the energy can be roughly compared to the reactant energy) and rephrase the corresponding sentence in the manuscript along the lines of "this scan indicates that a concerted mechanism is less favorable".

Sincerely,
Andreas Goetz

Reviewer #4 (Remarks to the Author):

The authors have adequately addressed my comments through their revisions to the text, and have made a number of other improvements based on the advice of the other reviewers. These results are likely to be of great interest to protein engineers and biochemists. I therefore recommend that this manuscript be published without further revisions.

Response to reviewers' 2nd comments

Reviewer #1:

Comment (1):

Interesting results with important increased on activity on the new created lipase in reactions in organic solvents. This open another question, which is the stability tolerance of this new artificial enzyme compared to the wild type??

Response (1):

Thank you very much for your important comments.

Actually, CALB is relatively stable in (hostile) organic solvents, and many papers about the enzymatic transesterification or promiscuous reactions catalyzed by WT-CALB or mutants in various organic solvents have been reported so far. Thus we think that WT CALB and QW4 variant display the different activities toward the transesterification of the model substrate (**1**) using methanol, probably due to their different structure of the active sites. It is similar to the difference between their activities toward the model hydrolysis of substrate (**1**), which has been explained in the original main text (for example, see the 3rd paragraph in the section of "Discussion and conclusions", page 8).

Comment (2):

New examples on improvement or maintaining the selectivity has been included, for example with the substrate **21**, but in the table no ee (or E value) has been shown although HPLC shows a only one enantiomer. This result must be commented on the text in the main manuscript, it was not mention. The conversion is low, is was possible to get higher conversion to evaluate the selectivity??

Response (2):

Thank you very much for your important comments.

We have added a short discussion about the hydrolysis of substrate **21** catalyzed by QW4 and QW10 (Line 5-10, right column, page 5).

According to the E value equation, the E values of the reactions of *trans*-**21** catalyzed by QW4 and Qw10 (Supplementary Table S3, entry 21-22) are >200. However it is meaningless to some extent because the conversions are lower than 5%. Thus we did not provide the E values for QW4 and QW10 (Supplementary Table S3, entry 21-22).

Indeed, the conversions of *trans*-**21** under the catalysis of QW4 and QW10 are much lower than that of WT and QW1, probably due to the reshaped active sites of QW4 and QW10. We also prolonged the reaction time, and the conversions cannot be significantly improved. Extensive optimization of the reaction conditions would be necessary to get reliable data for enantioselectivity, but this is not the object of our mechanistic study.

Comment (3):

In HPLC profile for the hydrolysis of **15**, differences on the retention time of the enantiomers between the sample and the results with the catalysts can be observed (FigS18) From 12-14 min for sample 20-26 for the rest, why??? The experimental conditions were different??? An explanation must be included.

Response (3):

Thank you very much for your careful checking and good suggestion.

In Supplementary Figure S18, the two peaks ($T_R = 11.8$ min, $T_S = 12.8$ min) in Fig. S18(a) are the R- and S-enantiomers of **15** (the ester substrate). The peaks in Fig. S18(b)-(e) are the S- and R-enantiomers of **16** (the alcohol product). They are different compounds in Fig. S18(a) and Fig. S18(b)-(e).

You are right, the retention times of **16** (the alcohol products) in Fig. S18(b)-(e) are slightly different (< 1 min) because of some unexpected small changes of HPLC conditions during the analysis process, such as the temperature, experimental time and the apparatus stability and repeatability through experiments. Now, we have repeated the HPLC analysis of the standard sample of **16** (the alcohol product), and the retention time of the standard **16** in Fig. S18(b) is close to that of the reaction products in Fig. S18(c)-(e).

Reviewer #2:**Comment (1):**

All my points have been fully addressed. I strongly support the publication of this work.

Response (1):

Thank you very much for your encouraging comment.

Reviewer #3:**Comment (1):**

The authors have addressed most of my concerns. This article is publishable after checking and addressing my comment on the reaction mechanism below.

Response (1):

We would like to thank the referee for his/her positive comment.

Comment (2):

In particular, new data indicates that, contrary to QW10, for QW4 a 2-step mechanism is favorable over a concerted mechanism. However, I would suggest to reformulate the sentence on page 8, right column, top ("However, no transition state could be located, indicating the improbability of a concerted mechanism for variant QW4"). This statement seems contradictory to the 2D scan presented in Figure S13, which seems to indicate that there is a transition state. More appropriate would be state that the scan indicated that a concerted mechanism is less favorable.

I am attaching an annotated version of the 2D potential energy surface scan from Figure S13. I indicated what I understand to be the position of the reactant, TS1, IM1, TS2 and product for the 2-step mechanism. I would like to ask the authors to label these points in their Figure, as it is hard to understand otherwise what is going on.

Response (2):

We appreciate the referee's recognition that a 2-step mechanism is the favored process for QW4 and not a concerted mechanism.

We appreciate the referee's advice. The point of the referee has been taken: we have labeled the possible TS1, IM1, TS2, and the product for the 2-step mechanism in Figure S13 in the revised

manuscript (Supplementary Figure S13, page S21). We have also labeled the point which is closest to the potential transition state corresponding to the concerted mechanism. However, optimization of the structure still leads to the deprotonated Cys105. It is also worth noting that the concerted mechanism requires that the proton does not transfer from Cys105 to His224 prior to the nucleophilic attack of Cys105 sulphur to the substrate carbonyl, indicating that the concerted mechanism is not favorable.

Taking the point of the referee, we have rephrased the statement on page 8 "However, no transition state could be located, indicating the improbability of a concerted mechanism for variant QW4" to "However, optimization of the point which is closest to the transition state corresponding to the concerted mechanism always led to the Cys105-/His224+ zwitterionic pair, indicating that the concerted mechanism is not preferred for variant QW4." in the revised manuscript (line 3-7, right column, page 8).

Comment (3):

Unfortunately all stationary points are outside of the scan region. The reactant is at a C105(S)-C(substrate) distance of around 3.7 Angstrom (inferred from QM/MM data in Figure R3) and should be lower in energy than the zero point of the 2D scan. Similarly, TS1 should be at a lower energy. Similarly, IM1 will be at a C105(SH)-H224(Ne) distance shorter than 1.4 Angstrom, the same for the product, which will also be at a C(105)S-C(substrate) distance of around 1.8 Angstrom (the 2D scan stops at 2.0). It is thus impossible to infer quantitative data about the energetics from the plot.

Response (3):

We thank the referee for his/her careful examination of the data.

In the study of the reaction mechanism of QW4, we actually first explored the possibility of concerted mechanism. However, no transition state corresponding to the concerted mechanism could be located (scan 1, Supplementary Figure S13b). We then explored the concerted mechanism starting from a different starting structure taken from the MD simulation (scan 2, Figure 5b, Supplementary Figure S12b). Again, no transition state could be located. Interestingly, a zwitterionic ionic pair of deprotonated Cys105 and protonated His224 was observed (IM1), from which we conducted forward and reverse scans to achieve the structures of the reactant and product (scan 2, Figure 5b, Supplementary Figure S12b). Since the starting structures for these two scans are different, the distance between C(105)S-C(substrate) in scan 1 (2.70 Å), is different from that of the starting structure in scan 2 (4.19 Å using B3LYP method and 3.74 Å using wB97x-D method in QM/MM optimizations).

Reactant in scan 1

Reactant in scan 2

Figure R1. Reactant structures used for potential energy surface scan 1 (Figure S13b) and scan 2 (Figure 5b, Figure S12). Both structures were optimized using QM/MM method with the QM region calculated by B3LYP/6-31+G(d,p) or ω B97x-D/6-31G+(d,p), respectively.

Comment (4):

However, unless I misunderstand the reaction coordinates and location of stationary points, it looks to me like there is a transition state for a concerted reaction, as indicated in the attached figure. It is probably higher in energy than TS1 for the 2-step mechanism, but it would be wrong to say that such a transition state does not exist.

The authors should check this (I am not asking for an optimization of the transition state but the energy can be roughly compared to the reactant energy) and rephrase the corresponding sentence in the manuscript along the lines of "this scan indicates that a concerted mechanism is less favorable".

Response (4):

As mentioned above, we examined the possibility of the concerted mechanism by 2D scans. However, optimization of the point which is closest to the transition state corresponding to the concerted mechanism always led to the Cys105-/His224+ zwitterionic pair, indicating that the concerted mechanism is not preferred. Following the referee's suggestion, we have labeled the potential stationary points associated with the two possible mechanisms in the revised manuscript (Supplementary Figure S13, Page S21).

We would like to thank the referee for the advice. Taking the point of the referee, we have rephrased the statement on page 8 "However, no transition state could be located, indicating the improbability of a concerted mechanism for variant QW4" into "However, optimization of the point which is closest to the transition state corresponding to the concerted mechanism also led to the Cys105-/His224+ zwitterionic pair, indicating that the concerted mechanism is not preferred for variant QW4." in the revised manuscript (Line 3-7, right column, page 8).

Reviewer #4:

Comment (1):

The authors have adequately addressed my comments through their revisions to the text, and have made a number of other improvements based on the advice of the other reviewers. These results are likely to be of great interest to protein engineers and biochemists. I therefore recommend that this manuscript be published without further revisions.

Response (1):

Thank you very much for your encouraging comment.

Statement about other revisions:

1. The title of the subheading “Altered reaction mechanism of cysteine-CALB variant QW4 unveiled by QM/MM calculations” in Page 8 was shorten to “Altered reaction mechanism of cysteine-CALB variant QW4”, less than 60 characters (incl spaces) according to the format requirements of Nature Communications.

2. Figure legends and Tables were placed at the end of the maintext (page 12-14) according to the section order in the format checklist.

REVIEWERS' COMMENTS:

Reviewer #1 (Remarks to the Author):

Authors have modified and give explanation to the last comments in this new revised version and in my opinion the manuscript now is suitable for publication.

Reviewer #3 (Remarks to the Author):

The authors have addressed my concerns through revision of the text. The statements regarding the reaction mechanism are now sufficiently careful as to not overstate results from limited potential energy scans in a complex high-dimensional system. I have no further comments.